# Structure, function and substrate preferences of archaeal *S*-adenosyl-ʟ-homocysteine hydrolases
Lars-Hendrik Koeppl [1,2], Désirée Popadić [1,2], Raspudin Saleem-Batcha [1,2], Philipp Germer[1] & Jennifer N. Andexer [1] ✉

*S*-Adenosyl-ʟ-homocysteine hydrolase (SAHH) reversibly cleaves *S*-adenosyl-ʟ-homocysteine, the product of *S*-adenosyl-ʟ-methionine-dependent methylation reactions. The conversion of *S*-adenosyl-ʟ-homocysteine into adenosine and ʟ-homocysteine plays an important role in the regulation of the methyl cycle. An alternative metabolic route for *S*-adenosyl-ʟ-methionine regeneration in the extremophiles *Methanocaldococcus jannaschii* and *Thermotoga maritima* has been identified, featuring the deamination of *S*-adenosyl-ʟ-homocysteine to *S*-inosyl-ʟ-homocysteine. Herein, we report the structural characterisation of different archaeal SAHHs together with a biochemical analysis of various SAHHs from all three domains of life. Homologues deriving from the Euryarchaeota phylum show a higher conversion rate with *S*-inosyl-ʟ-homocysteine compared to *S*-adenosyl-ʟ-homocysteine. Crystal structures of SAHH originating from *Pyrococcus furiosus* in complex with S$_L$H and inosine as ligands, show architectural flexibility in the active site and offer deeper insights into the binding mode of hypoxanthine-containing substrates. Altogether, the findings of our study support the understanding of an alternative metabolic route for *S*-adenosyl-ʟ-methionine and offer insights into the evolutionary progression and diversification of SAHHs involved in methyl and purine salvage pathways.

The methylation of small molecules as well as nucleic acids and proteins is an important modification in nature found in various biological processes such as drug metabolism, epigenetic regulation, and cancer development[1–3]. The methyl groups for such modifications are provided by a global metabolic pathway, the methyl cycle. The enzyme cofactor needed for this reaction is *S*-adenosyl-ʟ-methionine (SAM), which is converted to *S*-adenosyl-ʟ-homocysteine (SAH) by methyltransferases (MTs; EC 2.1.1.x) once the methyl group is transferred to a substrate, e.g. DNA, RNA, proteins, or small molecules[4,5]. MTs are a diverse group of enzymes installing the methyl group onto O, N, S, and C atoms, among others. In cells, the by-product SAH is salvaged as part of a complex regulation system, as SAH is known to act as a negative feedback inhibitor on most MTs, reducing the methylation rate in the cell[6].

The SAM/SAH ratio is controlled by different enzymatic pathways (Fig. 1). 5′-Methylthioadenosine (MTA)/SAH nucleosidase (MTAN; EC 3.2.2.9) irreversibly cleaves the glycosidic bond of SAH to adenine and *S*-ribosyl-ʟ-homocysteine, which is further transformed to ʟ-homocysteine

(Hcy) by the *S*-ribosyl-ʟ-homocysteine lyase LuxS (EC 4.4.1.21)[7]. Alternatively, SAH hydrolase (SAHH; EC 3.3.1.1) reversibly converts SAH to adenosine and Hcy[7,8]. Hcy is re-methylated to ʟ-methionine, a building block for SAM formation, by different enzymes[9–11].

Some organisms feature both routes (MTAN/LuxS and SAHH) for SAH degradation, while others only possess one, or in rare cases, e.g. in *Mycoplasma genitalium*, neither pathway[12]. A third pathway was identified in the archaeon *Methanocaldococcus jannaschii*. A 5′-deoxyadenosine deaminase (DadD; EC 3.5.4.41) catalyses the deamination of SAH to *S*-inosyl-ʟ-homocysteine (SIH)[13], which is subsequently cleaved to inosine (Ino) and Hcy by the SAHH homologue from *M. jannaschii*. For SAH, no activity was detected and therefore the enzyme was reclassified as an SIH hydrolase (SIHH; EC 3.13.1.9)[14].

A homologue of DadD, an MTA/SAH deaminase (EC 3.5.4.31/28), was found in *Thermotoga maritima* in a structure-based activity prediction in 2007, indicating the same metabolic pathway as in the archaeon *M. jannaschii*. The SAHH from this thermophilic bacterium was shown to

[1]Institute of Pharmaceutical Sciences, University of Freiburg, Albertstr. 25, 79104 Freiburg, Germany. [2]These authors contributed equally: Lars-Hendrik Koeppl, Désirée Popadić, Raspudin Saleem-Batcha. ✉e-mail: jennifer.andexer@pharmazie.uni-freiburg.de

**Fig. 1 | SAH degradation pathways.** SAH is either reversibly cleaved to adenosine and Hcy by SAHH (green) or degraded in two steps catalysed by MTAN and LuxS (orange). The pathway discovered in *Methanocaldococcus jannaschii* contains the deamination step of SAH to SIH, which is subsequently cleaved to inosine and Hcy by SIHH (blue).

catalyse SIH cleavage in addition to SAH cleavage (with $K_M$ values in the same order of magnitude)[15]. In 1988, a putative homologue of the deaminase was identified in *Streptomyces flocculus* (*Streptomyces albus* ATCC 13257) and proposed to be part of a major route in SAH metabolism, as SIH was isolated from the organism[16,17]. In addition to the SAHH homologues from *M. jannaschii*[14] and *T. maritima*[15] that were tested for SIH cleavage or synthesis activity, one bacterial representative from *Alcaligenes faecalis* was tested with nucleoside analogues in the synthesis reaction in 1984. With inosine, the enzyme showed almost no activity (0.5%) compared to adenosine as substrate[18].

SAHHs are highly conserved in all domains of life and have been extensively characterised biochemically, as well as structurally, over the last century[19,20]. As the products of the SAHH-catalysed cleavage are rapidly removed in vivo, this reaction is preferred in cells, while SAH synthesis is the main reaction taking place in vitro[21,22]. SAHH depends on the cofactor nicotinamide adenine dinucleotide in its oxidised form (NAD$^+$), which is self-regenerated during the catalytic cycle[23–26]. Briefly, the 3'-OH group of the adenosine ribose is oxidised to a ketone by reducing NAD$^+$ to NADH. The now more acidic C4' proton is abstracted to form a carbanion intermediate, followed by the release of Hcy. Water is added through a Michael-type addition to the C4'–C5' double bond[23]. The last step is the reduction of the ketone to form adenosine under re-oxidation of NADH to NAD$^+$; thus, the cofactor is ready for the next reaction cycle either in the cleavage or synthesis direction (Supplementary Fig. S25 online).

The protein structure of SAHHs features three domains in a monomer, the active form is usually a homotetramer[19] (Fig. 2a, b). The substrate-binding domain is located next to the cofactor-binding domain, each showing a Rossmann fold[19]. The C-terminus is a smaller dimerisation domain. The substrate-binding and the cofactor-binding domains are connected by a two-part hinge element. SAHHs have been shown to alternate between two conformations differing in the relative positions of the substrate- and cofactor-binding domains (Fig. 2c). In the "open" conformation, with no substrate bound, the domains are away from each other providing access to the active site (PDB IDs: 3X2F and 4LVC)[27,28]. In the "closed" conformation, the substrate-binding domain reorients by about 18° relative to the cofactor-binding domain and both domains form the active site interacting with a substrate or inhibitor[29,30]. In addition to the overall conformational state, a critical loop region comprises a pair of histidine

(His) and phenylalanine (Phe) acting as gate residues that provide a channel for the substrate to access the active site; it is highly conserved over SAHH sequences of all domains of life[31]. This so called "molecular gate" loop displays a plasticity that opens (His-OUT) and shuts (His-IN) upon different ligation states *via* a 180° flip of the peptide plane between Cα atoms of His and Phe (Fig. 2c)[31].

Multiple protein structures from bacterial and eukaryotic SAHH enzymes have been determined[26,28,31,32] (Supplementary Table S4 online); however, no protein structure originating from the domain of Archaea has been published. The archaeal enzymes are predicted to show similar architecture as their eukaryotic and bacterial relatives but display remote differences such as a shortened C-terminus and a missing 40 amino acid segment in the catalytic domain among other smaller deletions[12,33]. Further, an HxTxQ(E) sequence signature (with x corresponding to any amino acid) is found in the substrate-binding domain of eukaryotic and bacterial enzymes, while extremophile SAHHs have been suggested to feature a different motif, HxT(E)xK[19]. The His residue is important for catalysis (Supplementary Fig. S25 online), while Thr(Glu) and Gln(Glu) have been suggested to stabilise the nucleobase in the active site *via* hydrogen bonds[19,25]. The residue Gln(Glu) was also found to play a role in the conformational changes of the enzyme, regulating its activity[34].

The discovery of SIHHs[14] prompted us to study the tertiary structures and biochemical properties of SAHH enzymes from archaea and to identify their differences from bacterial and eukaryotic homologues characterised biochemically. To the best of our knowledge, we show here the first archaeal SAHH/SIHH crystal structures within the Euryarchaeota and Crenarchaeota phyla in the presence of hypoxanthine derivatives (inosine and SIH) of the recently proposed alternate SAM salvage pathway[13]. In addition, our investigation of the substrate ranges of different SAHHs led to the identification of more organisms potentially featuring alternative SAM regeneration pathways. Altogether these findings result in a substantial support of the alternate SAM salvage pathway and the importance of methyl metabolism in archaea.

## Results and discussion
### Overview of investigated SAHHs/SIHHs
In this work, SAHHs/SIHHs from archaea, bacteria, and eukaryotes were characterised biochemically (Supplementary Table S2 online). The

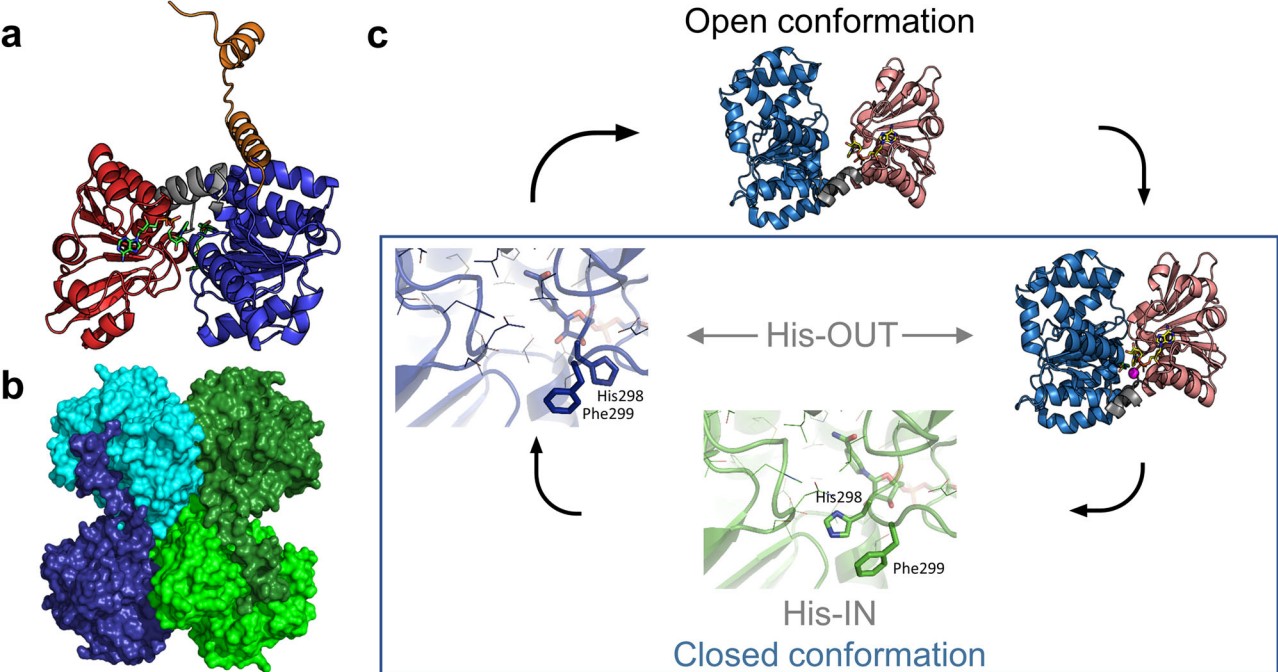

**Fig. 2 | Structural features of SAHHs. a** An enzyme monomer consists of the C-terminal dimerisation domain (orange), the substrate-binding domain (blue) and the cofactor-binding domain (red) connected by a two-part hinge element (grey) (PDB ID: 7R38). **b** The monomers oligomerise to form a dimer of a dimer (PDB ID:

7R38). **c** The monomers alter their conformation between the open and closed state (PDB ID: 4LVC). Within the closed conformation, a pair of His and Phe can act as gate residues (PDB IDs: 7R37 and 7R38).

homologues from *M. jannaschii* (*Mj*SIHH) and *T. maritima* (*Tm*SAHH) were previously reported to have SIH cleavage activity[14,15,27,35]. Structures of archaeal SAHHs were determined from three organisms: *Methanococcus maripaludis* (*Mma*SAHH), *Pyrococcus furiosus* (*Pfu*SAHH)[36], and *Sulfolobus acidocaldarius* (*Sac*SAHH). In addition, the structure of SAHH from *Mus musculus* (*Mm*SAHH) was determined in complex with inosine.

### The substrate range differs depending on the origin of the enzyme

All enzymes were tested in both reaction directions with the hypoxanthine- and adenine-containing substrates. As SAH/SIH cleavage is not preferred in vitro and consequently hard to follow, the reaction was coupled to Hcy *S*-methylation catalysed by Hcy *S*-MT (HSMT; EC 2.1.1.10)[9] to drive the reaction forward by removing one of the products (Fig. 3b–d). The following analysis with HPLC-DAD allows the direct identification of hypoxanthine- vs. adenine-containing substrate, which is not possible in previously described coupled colorimetric and photometric assays for SAHHs[37,38]. As a side-product, the corresponding nucleobase (adenine or hypoxanthine) was detected in all set-ups, as previously described[39] (Fig. 3b, c). In some reports, SAHH activity was only observed with externally added NAD$^{+}$[40–42], while there are also examples showing SAH cleavage and synthesis activity without the addition of extra NAD$^{+}$[32,36,43]. All enzymes tested in this work were active without the addition of NAD$^{+}$.

Under the chosen conditions, all SAHHs/SIHHs were active for SAH cleavage and synthesis (Table 1), including the homologue from *M. jannaschii* (*Mj*SIHH) previously described to be specific for SIH as substrate[14] (Supplementary Fig. S10; all chromatograms in Figs. S4–S21 online). Nevertheless, this enzyme clearly prefers the hypoxanthine-containing compounds over the adenine-containing ones. The same results were observed for other representatives from Euryarchaeota (*Mi*SAHH, *Mma*SAHH, *Pfu*SAHH, and *Tk*SAHH). As described before, the bacterial *Tm*SAHH, which is closely related to these euryarchaeal enzymes (Fig. 3a), catalysed SAH and SIH cleavage and synthesis[15], in our hands also with a strong preference for the hypoxanthine derivatives. In contrast, another subgroup of SAHHs from euryarchaeal origin (*Mc*SAHH,

*Me*SAHH, *Mh*SAHH, and *Mt*SAHH) showed a substantial preference for the synthesis and cleavage of SAH over SIH. In the case of *Me*SAHH, neither synthesis nor cleavage of SIH was catalysed while the other enzymes of this subset showed slight catalytic activity for SIH synthesis. Comparable results were obtained for the SAHHs from Crenarchaeota (*Sac*SAHH and *Sso*SAHH), which were only active with SAH or adenosine as substrates. These findings strongly support the assumption that alternative routes for methyl metabolism coupled to purine salvage exist within some classes of Euryarchaeota and closely related bacteria.

In previous work, we successfully used *Mm*SAHH for an in vitro SAM regeneration cycle with alternative nucleobases including hypoxanthine[44]. This activity was now confirmed in the individual cleavage and synthesis reactions; however, reactions with SAH were in this case preferred. A similar pattern was observed for the SAHH from the bacterium *Pseudomonas aeruginosa*, while neither the representative from *Corynebacterium glutamicum* nor from the plant *Lupinus luteus* accepted inosine or SIH as substrates. As SIH had been described as a metabolite in *S. flocculus*, along with the presence of an SAH deaminase[16,17], we tested SAHHs from two *Streptomyces* species (*Sa*SAHH and *Sf*SAHH). In both cases a low activity for SIH synthesis was detected; however, no activity for SIH cleavage (Supplementary Figs. S17 and S18 online). It might be that SIH is metabolised by other enzymes in these bacteria, suggesting an alternative SAH metabolism pathway than the one described for Euryarchaeota and archaeal-type bacteria.

### Increased temperatures do not influence the substrate preferences

So far, the substrate preferences of archaeal, bacterial, and eukaryotic SAHH/SIHH representatives were tested at 37 °C to investigate their applicability in a multi-enzyme SAM regeneration cycle[44]. Some of the archaeal (*Pfu*SAHH, *Tk*SAHH, *Mc*SAHH, *Mi*SAHH, *Mt*SAHH, *Mj*SIHH, *Sac*SAHH, and *Sso*SAHH) as well as archaeal-type (*Tm*SAHH) enzymes are derived from (hyper-)thermophilic organisms. *Tm*SAHH produced and purified at room temperature was previously described to require thermal activation to attain enzymatic activity[27,42]. However, in our hands *Tm*SAHH,

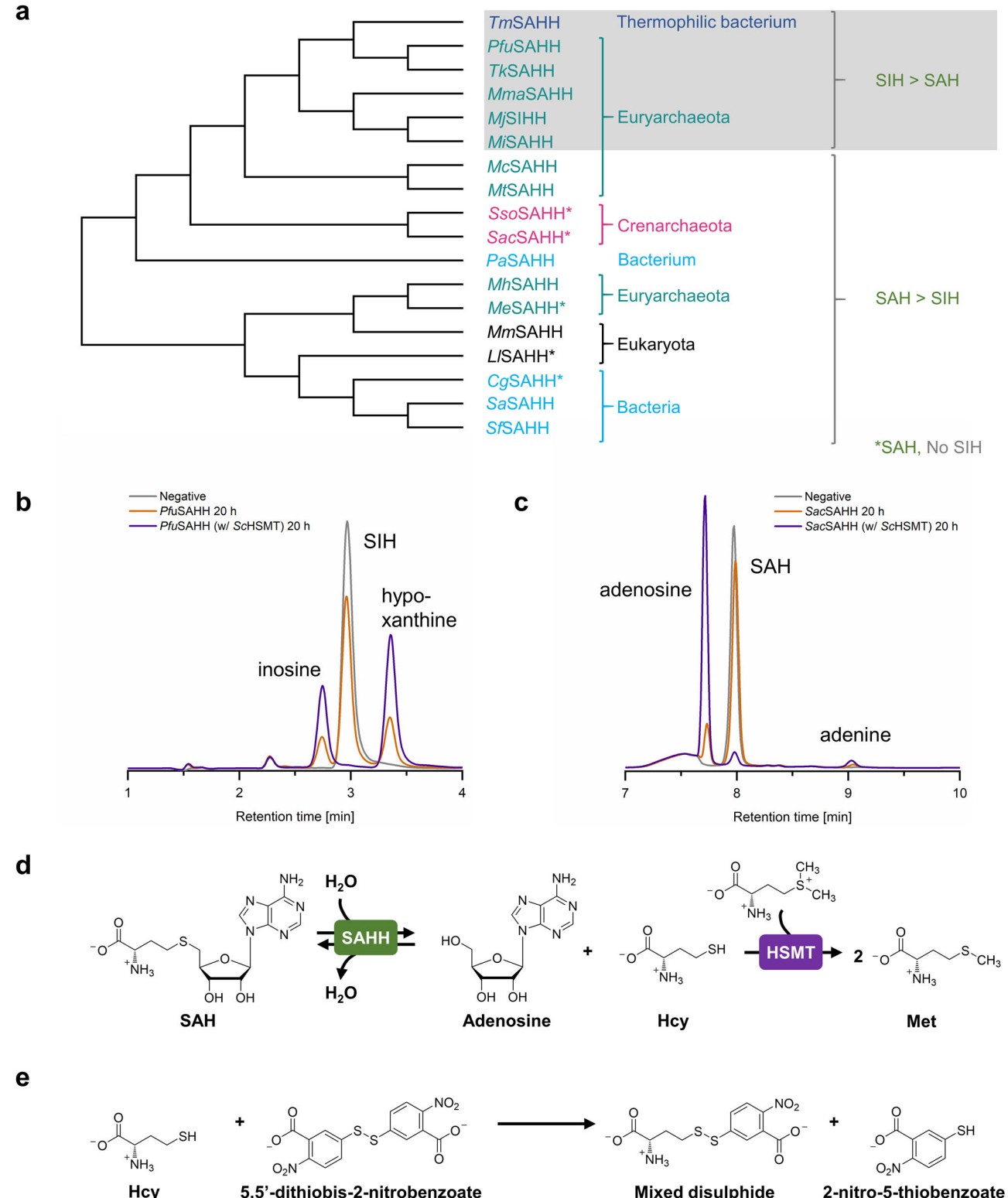

**Fig. 3 | Phylogenetic tree of investigated SAHHs/SIHHs with substrate range and used enzyme assays. a** The sequence-based phylogenetic tree groups the tested SAHHs/SIHHs according to their substrate preference. Nevertheless, this does not fully correspond to the phylogenetic relatedness of the respective organisms: Representatives from Crenarchaeota exclusively accept SAH as substrate; in contrast, most family members tested from Euryarchaeota and a closely related thermophilic bacterium accept both SAH and SIH, including some homologues that clearly prefer SIH. Except for *Cg*SAHH and *Ll*SAHH, bacterial and eukaryotic SAHHs accept both substrates with a substantial preference for SAH. **b** *Pfu*SAHH-catalysed cleavage of SIH, and (**c**) *Sac*SAHH-catalysed cleavage of SAH, both with and without addition of *Sc*HSMT (and negative control without enzymes). **d** The addition of the second enzyme *Sc*HSMT increases the conversion of the cleavage reaction. **e** The thiol scavenger 5,5'-dithiobis-2-nitrobenzoic acid (DTNB) reacts with Hcy to form a mixed disulphide and 2-nitro-thiobenzoic acid (TNB). For reactions performed at 70 °C it was used instead of *Sc*HSMT to increase the conversion of SAH/SIH cleavage.

**Table 1 | Overview of SAHHs biochemically tested in this work**

| Enzyme | Organism | Phylum/Kingdom | SAH Cleavage/Synthesis | SIH Cleavage/Synthesis | Figure SI online |
|---|---|---|---|---|---|
| *Sac*SAHH | *Sulfolobus acidocaldarius* | Crenarchaeota | +++/+++ | −/− | S4 |
| *Sso*SAHH | *Saccharolobus solfataricus* | Crenarchaeota | +++/+++ | −/− | S5 |
| *Mc*SAHH | *Methanocella conradii* | Euryarchaeota | +++/++ | −/+ | S6 |
| *Me*SAHH | *Methanohalobium evestigatum* | Euryarchaeota | +/++ | −/− | S7 |
| *Mh*SAHH | *Methanohalophilus halophilus* | Euryarchaeota | ++/++ | −/+ | S8 |
| *Mi*SAHH | *Methanocaldococcus infernus* | Euryarchaeota | +/+ | ++/++ | S9 |
| *Mj*SIHH | *Methanocaldococcus jannaschii* | Euryarchaeota | +/+ | +++/+++ | S10 |
| *Mma*SAHH | *Methanococcus maripaludis* | Euryarchaeota | +/+ | +++/+++ | S11 |
| *Mt*SAHH | *Methanothrix thermoacetophila* | Euryarchaeota | +++/++ | −/+ | S12 |
| *Pfu*SAHH | *Pyrococcus furiosus* | Euryarchaeota | +/++ | +++/+++ | S13 |
| *Tk*SAHH | *Thermococcus kodakarensis* | Euryarchaeota | +/+ | +++/+++ | S14 |
| *Cg*SAHH | *Corynebacterium glutamicum* | Actinomycetota | +++/+++ | −/− | S15 |
| *Pa*SAHH | *Pseudomonas aeruginosa* | Pseudomonadota | +++/+++ | +/++ | S16 |
| *Sa*SAHH | *Streptomyces albus* | Actinomycetota | +++/+++ | −/+ | S17 |
| *Sf*SAHH | *Streptomyces flocculus* | Actinomycetota | +++/+++ | −/+ | S18 |
| *Tm*SAHH | *Thermotoga maritima* | Thermotogota | +/+ | +++/+++ | S19 |
| *Ll*SAHH | *Lupinus luteus* | Plantae | +/+++ | −/− | S20 |
| *Mm*SAHH | *Mus musculus* | Animalia | +++/+++ | +/+++ | S21 |

Conversions indicated for the cleavage are based on reactions coupled to *Sc*HSMT.
[+++]High conversion (>70%), [++]Medium conversion (30–70%), [+]Low conversion (< 30%), - no conversion.

as well as the other thermophilic enzymes, was enzymatically active without previous thermal activation. Assays to test the influence of higher temperatures on the substrate preferences were performed at 70 °C in SAH/SIH synthesis and cleavage directions with two selected model enzymes, *Pfu*SAHH (Euryarcheota) and *Sac*SAHH (Crenarchaeota). For reactions performed at 70 °C, HSMT from *Saccharomyces cerevisiae* (*Sc*HSMT) was replaced by 5,5'-dithiobis-2-nitrobenzoic acid (DTNB). It acts as thiol scavenger reacting with Hcy in stoichiometric amounts to form a mixed disulphide and 2-nitro-thiobenzoic acid (TNB), and therefore shifts the reaction equilibrium to the product side[37,45] (Fig. 3e). The suitability of DTNB as substitute for *Sc*HSMT was tested beforehand at 37 °C (Supplementary Fig. S22 online). At 70 °C, *Pfu*SAHH still showed a clear preference for SIH synthesis and cleavage compared to SAH (Supplementary Fig. S23 online). For *Sac*SAHH, the conversion with the unfavoured substrates was slightly higher compared to the reactions performed at 37 °C; nevertheless, the enzyme still showed a substantial preference for SAH synthesis and cleavage (Supplementary Fig. S24 online). This strongly indicates that the tested SAHHs/SIHHs show the same substrate preferences at both temperatures. Increased conversion of *Sac*SAHH-catalysed SIH cleavage suggests that they could degrade small amounts of SIH under higher living temperatures.

### Different motifs for binding the nucleobase of the substrate/product in the subgroups of archaea

The differences in substrate range among the investigated SAHHs/SIHHs prompted us to conduct a detailed sequential and structural comparison to identify the underlying molecular basis. A sequence alignment of the enzymes tested in this work (Supplementary Fig. S26 online), showed that *Mm*SAHH as well as *Pa*SAHH are missing a 40 amino acid segment in the catalytic domain as opposed to the other eukaryotic and bacterial representatives. All investigated archaeal SAHHs/SIHHs, except for *Me*SAHH and *Mh*SAHH, also lack this structure segment as described before[12,33]. Another sequence feature discriminating *Me*SAHH and *Mh*SAHH from the other archaeal and archaeal-type homologues is the length of their C-terminus. The majority of SAHHs/SIHHs analysed in this work containing a shortened C-terminus are (hyper-)thermophilic, confirming the

assumption[36] that the length of the C-terminus correlates with the thermophilicity of the enzymes[46]. Based on our sequence analyses, the fingerprint motif suggested to distinguish extremophile SAHHs from mesophilic enzymes[19] can be further specified: Crenarchaeota and a large part of Euryarchaeota feature HxTxE as a signature, this matches the sequence signature of bacterial and eukaryotic representatives [HxTxE(Q)]. A subgroup of the euryarchaeal and archaeal-type bacterial SAHHs, including six homologues analysed in this study have an HxExK motif (Fig. 4). Based on the data on euryarchaeal sequences available in the UniProt database, this subgroup encompasses mainly the *Methanococci*, *Thermococci*, *Methanobacteria*, *Hadesarchaea*, *Archaeoglobi* and *Methanosarcina*; while euryarchaeal enzymes from various other classes (e.g. *Methanocella* and *Methanophagales*) feature the HxTxE motif as found in Crenarchaeota; in further classes (e.g. *Methanomicrobia* and *Theionarchaea*), there is no clear trend visible. Our experiments show that the homologues with a HxExK motif prefer SIH as substrate while enzymes with a HxTxE(Q) motif prefer SAH. These results infer that the fingerprint motif distinguishes SAH-preferring SAHHs from SIH-preferring SIHHs.

### Genomes from Euryarchaeota and archaeal-type bacteria encode a deaminase

In order to get more insight in the potential alternative SAH salvage pathway, a BLASTP[47] search for a protein homologue of DadD from *M. jannaschii* (*Mj*DadD, Fig. 1) in the organisms producing the SAHHs/SIHHs analysed in this work was performed. The genomes from all analysed Euryarchaeota, as well as from the archaeal-type bacterium *T. maritima*, were found to have a homologue encoded (Supplementary Table S6 online). This is inconsistent with the substrate scope of characterised archaeal SAHHs/SIHHs as the subgroup of the tested euryarchaeal *Mc*SAHH, *Me*SAHH, *Mh*SAHH and *Mt*SAHH showed no activity for SIH cleavage. Notably, the BLASTP search revealed multiple isoforms of *Me*SAHH (UniProt accessions: D7E8L4 and D7E8N1) and *Mh*SAHH (A0A1L3PZV2) suggesting that those organisms contain more than one SAHH, with putatively different substrate preferences, as inferred from their fingerprint motif. For *Mc*SAHH and *Mt*SAHH, no isoforms were found. In these organisms, SIH resulting from DadD catalysed SAH deamination

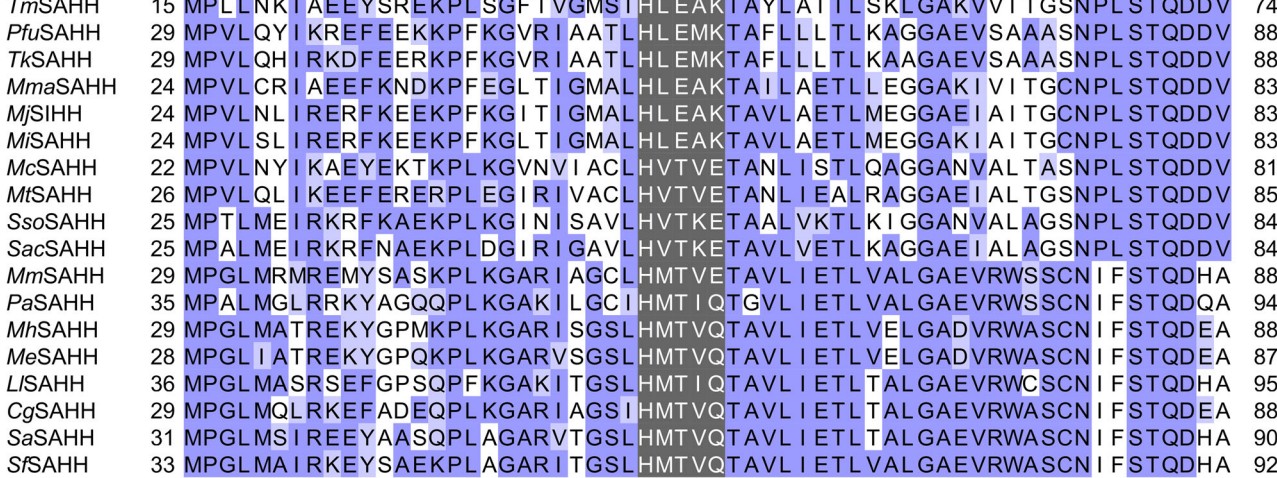

**Fig. 4 | Sequence analysis of nucleobase-binding motifs of SAHHs/SIHHs.** Partial alignment of the amino acid sequences of SAHHs/SIHHs tested in this study. The fingerprint motifs are highlighted in grey while the other residues are coloured by conservation according to the BLOSUM62 algorithm.

needs to be degraded *via* a pathway without SAHHs/SIHHs. Hence, a BLASTP search for homologues of *Escherichia coli* MTAN (*Ec*MTAN), cleaving SAH into adenine and *S*-ribosyl-ʟ-homocysteine, was performed. Only *C. glutamicum*, *T. maritima* and *L. luteus* were found to encode an MTAN homologue (Supplementary Table S7 online). This suggests that in the organisms encoding a DadD and an SAHH only cleaving SAH, SIH is metabolised by another pathway. Moreover, the lack of MTAN homologues in Euryarchaeota encoding an SAHH preferring SIH but also accepting SAH as substrate, strongly indicates that those enzymes have a dual function for SIH and SAH metabolism. The organisms without an encoded SAH deaminase do not need their SAHH to degrade SIH in addition to SAH. A more detailed BLASTP search including all available entries in the UniProt database[48] resulted in the identification of further homologues of *Mj*DadD, exclusively within the domain of bacteria and the phylum of Euryarchaeota. This suggests that the alternative SAM regeneration pathway *via* SIH is indeed not present in all archaea, but only in a subgroup from the phylum of Euryarchaeota and closely related thermophilic bacteria. Due to their shared living environment, thermophilic bacteria were shown to have obtained genes *via* horizontal transfer from archaea[49]. The reason for substantial SIH degradation and synthesis by *Mm*SAHH and *Pa*SAHH remains yet unclear. Likely, there will be other amino acid residues involved in substrate recognition which are not obvious from the sequence alignments.

## Architecture and domains of archaeal SAHHs

Thus far, the sequence comparison revealed differences concerning the constitution of the catalytic domain, the length of the C-terminus, and the composition of the signature motif binding the nucleobase moiety of substrates and products. A detailed structural analysis was performed to

### Table 2 | Summary of crystal structures solved in this study

| Enzyme | Substrate | Cofactor | Resolution | Treatment | PDB ID |
|---|---|---|---|---|---|
| *Pfu*SAHH | SIH | NAD⁺ | 2.0 Å | 22–25 °C | 7R38 |
| *Pfu*SAHH | Inosine | NAD⁺ | 2.3 Å | 22–25 °C | 7R37 |
| *Pfu*SAHH | Inosine | NAD⁺ | 2.0 Å | 95 °C, 15 min | 8QNO |
| *Mma*SAHH | Inosine | NAD⁺ | 2.5 Å | 22–25 °C | 7R3A |
| *Sac*SAHH | Adenosine | NAD⁺ | 2.6 Å | 22–25 °C | 7R39 |
| *Mm*SAHH | Inosine | NAD⁺ | 2.5 Å | 22–25 °C | 8COD |

The molecules in the asymmetric unit in all the cases were almost identical, being superimposable with a root-mean-square deviation (r.m.s.d.) of 0.17–0.63 Å over 356–393 Cα atoms. Data collection and refinement statistics are summarised in Table 3.

determine how those differences influence the three-dimensional structure of SAHHs/ SIHHs and effect the substrate preferences. The archaeal enzymes were an ideal model system as they show different substrate preferences; however, no structures of archaeal SAHHs have been published to date. Here, we present the structures of three archaeal SAHHs (*Pfu*SAHH, *Mma*SAHH, *Sac*SAHH); in addition, the structure of the eukaryotic *Mm*SAHH was determined in complex with its unusual substrate inosine (Table 2). In all SAHH structures obtained, the NAD⁺ cofactor was present indicating that it was co-purified with the enzyme.

The archaeal SAHH monomer contains the same three domains as previously studied SAHHs across other domains of life: substrate-binding domain, cofactor-binding domain, and C-terminal domain[27,32,34,42,50]. The binding modes of NAD⁺ at the cofactor-binding domain and inosine or adenosine in the substrate-binding domain of archaeal SAHHs are similar to those observed in other SAHHs of distinct origin (Fig. 5a–d)[26,28,31,32,34,51,52]. In this study, the nucleoside moieties of the three ligands, adenosine, inosine, and SIH, participate in a similar pattern of hydrogen-bonding and non-bonding interactions (Supplementary Table S5 online). Evaluating the euryarchaeal structures elucidated in this study, it supports the assumption that the Glu57 residue (in the HxExK motif, numbering according to *Pfu*SAHH, Fig. 5c/d and S27A/B online) forms a hydrogen bond with the heterocyclic nitrogen atom N1 of the nucleoside, analogous to the Thr residue in crenarchaeal, bacterial and eukaryotic homologues (in the HxTxE motif, e.g. Thr53 in *Sac*SAHH, Fig. 5a and S27D online). The Lys residue at the end of the motif is found in enzymes that prefer hypoxanthine-containing substrates; here, the positively charged side chain can form a hydrogen bond with the oxygen at C6 in inosine, as seen in the structures of *Pfu*SAHH (Lys59 in Fig. 5e and S27A/B online) and *Mma*SAHH (Lys74 in Supplementary Fig. S27C online), contributing to the stabilisation of the substrate in the active site. The motif HxTxE found in *Sac*SAHH (Crenarchaeota) shows the same interactions with the substrate adenosine as in bacterial and eukaryotic enzymes[19]. As described before[19], we observed that the *exo*-amino group of adenosine bound in *Sac*SAHH forms additional hydrogen bonds with the carbonyl oxygen atoms in the main chains of Glu342 and His344 indicating that the adenine ring is in its preferred tautomeric amino form. Regarding the structures of *Pfu*SAHH and *Mma*SAHH with their preferred substrates inosine or SIH, the oxygen attached to C6 seems to form additional hydrogen bonds with the carbonyl oxygen atoms of Asp348 (*Pfu*SAHH) or Asp362 (*Mma*SAHH) in the main chains. This suggests that the hypoxanthine ring is in the imino-hydroxy form (Fig. 5f), although inosine has been shown to be most stable in the amino-oxo form in water[53]. Analysis of superposed structures of *Mm*SAHH with bound adenosine[54] and bound inosine showed no substantial

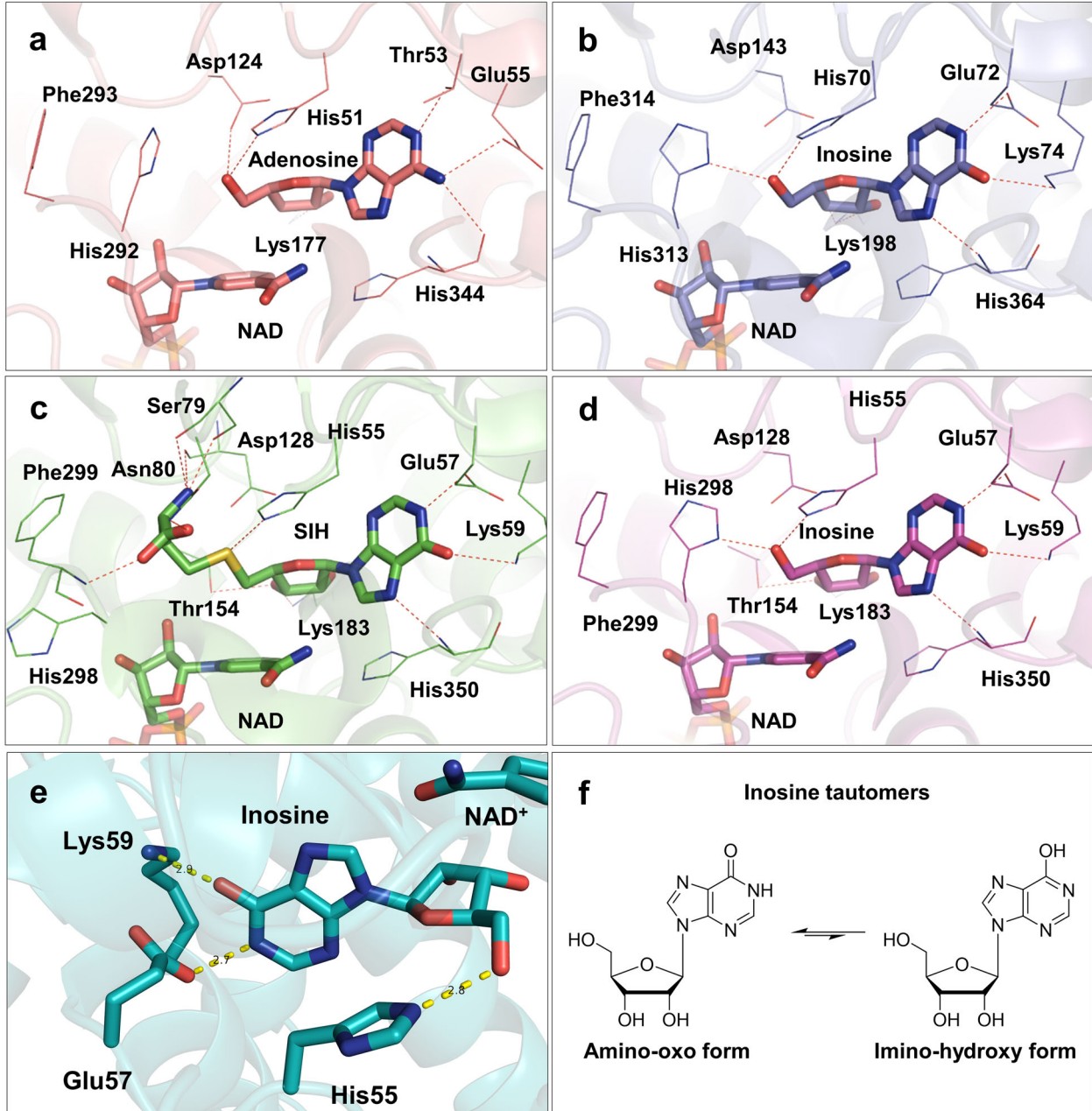

**Fig. 5 | Binding modes of substrates/products and the cofactor in archaeal SAHHs. a** Mode of adenosine and NAD$^+$ cofactor binding in the active site of *Sac*SAHH (PDB ID: 7R39). **b** Mode of inosine and NAD$^+$ cofactor binding in the active site of *Mma*SAHH (PDB ID: 7R3A). **c** Mode of SIH and NAD$^+$ cofactor binding in the active site of *Pfu*SAHH (PDB ID: 7R38). **d** Mode of inosine and NAD$^+$ cofactor binding in the active site of *Pfu*SAHH (PDB ID: 7R37). **e** Inosine is bound by the motif HxExK in the substrate-binding domain (cartoon representation; PDB ID: 7R37). **f** Tautomers of inosine. Ligands are represented as sticks.

differences in the architecture of the substrate binding pocket depending on the bound substrate (Supplementary Fig. S28 online).

## Overall conformational states and the molecular gate of archaeal SAHHs

Different conformational states of SAHHs have been described, where the open conformation was observed with no bound substrate/product in the substrate-binding domain, while the protein with a bound substrate or analogue shows the overall closed conformation. An example is a structure (PDB ID: 4LVC) of a bacterial SAHH showing adenosine bound in three subunits displaying a closed conformation while the fourth, ligand-free, subunit is in an open conformation[28]. All archaeal SAHH complexes in

this study also display a closed conformational state including the *Pfu*SAHH complex with SIH bound as substrate.

In addition to the closed overall conformation of the *Pfu*SAHH•NAD•SIH complex, the molecular gate loop displays a flexibility. The transitions between the overall conformation and the conformation of the gatekeeper residues are independent from each other. Yet, His298 and Phe299 forming the molecular gate in *Pfu*SAHH can only act as gatekeepers in the closed conformation when substrate-binding domain and cofactor-binding domain form the channel, as described before[28,31]. Interestingly, the His298 residue alone displays a noticeable conformational difference in this study. In the inosine bound complex, His298 is situated within the active site staging a His-IN conformation (gate shut) where its imidazole ring is

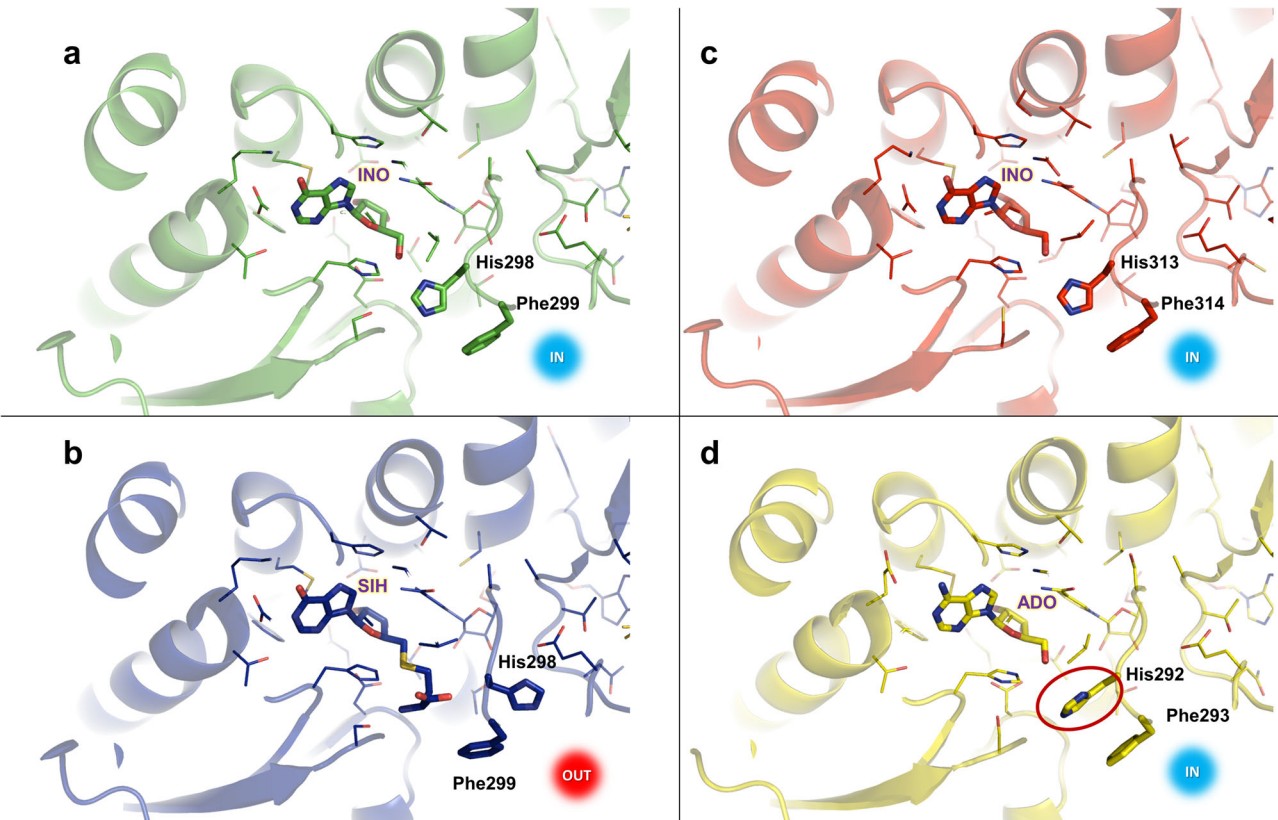

**Fig. 6 | The molecular gate residues in the crystal structures of archaeal SAHHs from this study.** The His residue states are represented as His-IN (IN) and His-OUT (OUT). **a** In the *Pfu*SAHH•NAD•inosine complex (PDB ID: 7R37), the His gatekeeper residue is in the IN orientation thereby closing the channel entrance. **b** In the *Pfu*SAHH•NAD•SIH complex (PDB ID: 7R38), the gatekeeper residue forms an OUT orientation leaving the channel entrance open. **c** For the *Mma*SAHH•NAD•inosine complex (PDB ID: 7R3A) the gatekeeper residue is in the IN orientation. **d** In the *Sac*SAHH•NAD•adenosine complex (PDB ID: 7R39), the gatekeeper residue is in the IN orientation that makes the channel entrance shut and inaccessible. The side chain of the gatekeeper residue of *Sac*SAHH (circled red) shows another rotamer conformation than in *Pfu*SAHH and *Mma*SAHH.

oriented towards the γ-carboxylate group of Asp128 with a distance of 4.2 Å. The Nδ1 group of His298 is oriented to the O5′ group of inosine with a distance of 2.7 Å. A similar pattern is observed for *Mma*SAHH and *Sac*SAHH complexes. However, the rotamer conformation of His in the His-IN state of *Sac*SAHH deviates from the rest of the His-IN conformations among the archaeal SAHHs (Fig. 6) in this study. This indicates that there are not only different His orientations possible in the His-OUT state[55], but also in the His-IN state. Comparison of the *Mm*SAHH structures with bound adenosine and inosine (Supplementary Fig. S29 online) and the structure of *Sac*SAHH suggests that the rotamer conformation is not dependent on the bound nucleoside substrate. The His298 residue of *Pfu*SAHH•NAD•SIH complex swings away from Asp128 by 10.4 Å and forms a hydrogen bond with the carboxylate group of Glu302 with a distance of 2.7 Å signifying a His-OUT conformation leaving an open molecular gate. In a previously described structure of the *Lupinus luteus* SAHH[32], the channel is open (His-OUT) even though adenosine is bound, while in the SAHH structure from *Mycobacterium tuberculosis* the same trend is seen as for *Pfu*SAHH with the channel closed (His-IN) with adenosine bound and channel opened with SAH bound[31]. Similarly, our complexes of *Pfu*SAHH and *Mma*SAHH with inosine and the complex of *Sac*SAHH with adenosine show a closed conformation state of the molecular gate in their active site (Fig. 6). In accordance with Manszewski et al., comparison of our structures with already published SAHH structures did not lead to an apparent correlation between the conformation of the gate residues and the bound substrates[56]. Instead, the state of the reaction catalysed by SAHHs may determine the conformation of the gatekeeper residues to protect the unstable 3′-keto intermediates from exposure to the aqueous environment, as proposed by Yang et al[57].

## Differences in archaeal compared to bacterial and eukaryotic SAHHs

While the protein sequences of archaeal SAHHs/SIHHs are highly conserved (Supplementary Fig. S32 online), the substrate range differs within the distinct phyla of archaea. The *Pfu*SAHH•NAD•inosine and *Mm*SAHH•NAD•inosine complexes determined in this work were used for structural comparison with the bacterial *Pa*SAHH (PDB ID: 6F3N). Besides the different signature motifs binding the nucleobase moiety, the overall architecture of the macromolecular environment of active sites is comparable between those three enzymes (Fig. 7a). The *Pfu*SAHH•NAD•inosine complex lacks a monovalent cation (Na+ for *Mm*SAHH and K+ for *Pa*SAHH) coordinated by the hinge element in proximity to the nucleobase binding pocket[34,52] (Fig. 7b–d). The absence of a comparable monovalent cation was observed in all structures of archaeal enzymes. Therefore, the catalytic activity of archaeal enzymes might be independent from the presence of monovalent cations as previously seen for a cyanobacterial homologue from *Synechocystis* sp. PCC 6803[52].

Looking at the overall structure of archaeal SAHHs compared to their eukaryotic and bacterial counterparts, the major difference is the shortened C-terminus, which interacts with the cofactor bound to the neighbouring subunit. Due to the shortened C-terminus, the archaeal enzyme structures lack hydrogen bonds, which are formed between amino acids (Lys426 and Tyr430) of one subunit with the 2′- and 3′-hydroxy groups of the adenosine moiety and the pyrophosphate of NAD+ bound in the adjacent subunit, present in *Mm*SAHH and other mesophilic homologues. Instead of interactions between C-terminal domains of neighbouring monomers, the major forces for stabilisation are provided by aromatic and hydrophobic amino acid residues at the interfaces of the tetramers[36]. In addition, the interfaces

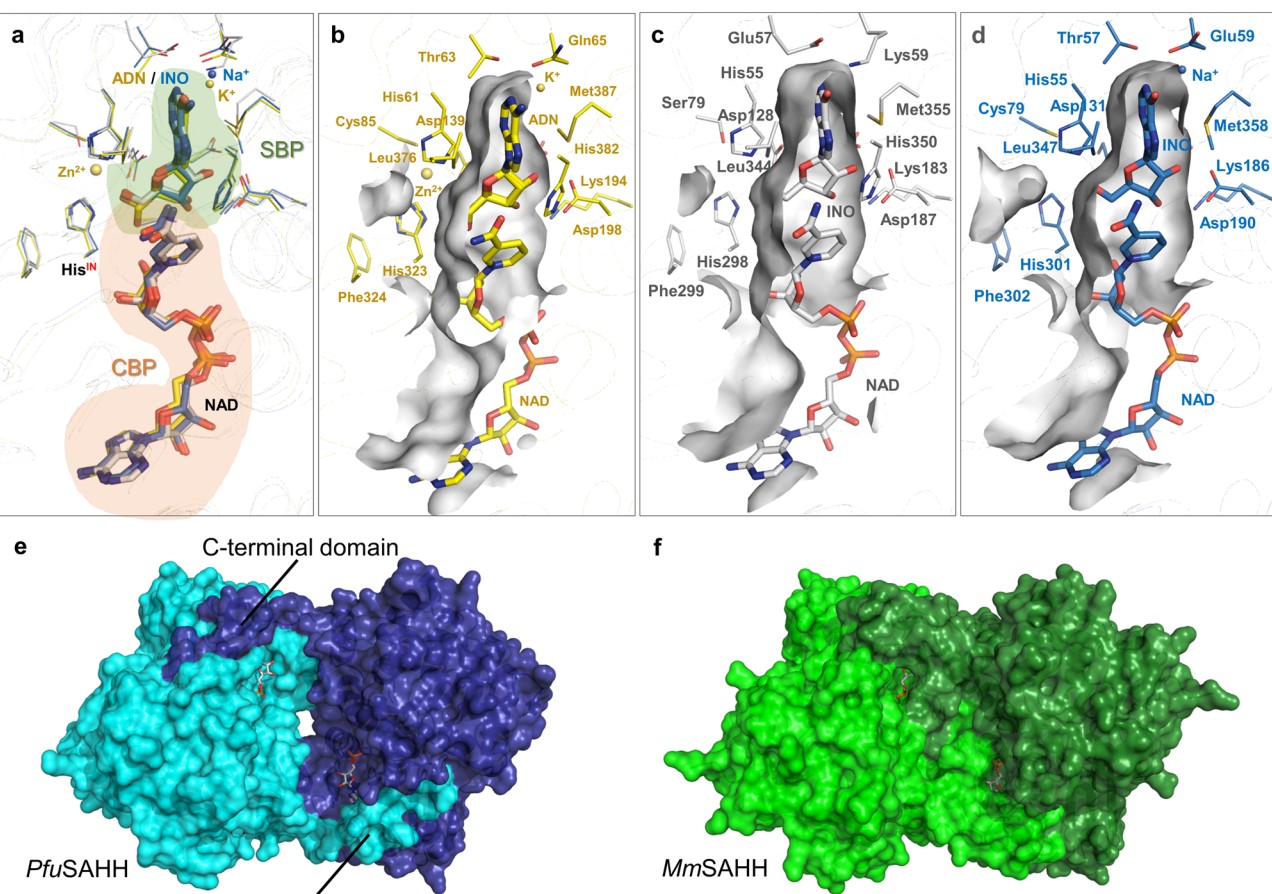

**Fig. 7 | Overall structural comparison of euryarchaeal *Pfu*SAHH with SAHHs from other domains. a** Superposition of the active sites of SAHHs from bacterial (yellow), eukaryotic (blue), and archaeal (white) organisms with bound substrates and NAD⁺ cofactors (SBP: substrate-binding pocket; CBP: cofactor-binding pocket). **b** SBP of bacterial *Pa*SAHH in complex with adenosine (ADN). $Zn^{2+}$ and $K^+$ are shown as yellow spheres (PDB ID: 6F3N[34]). **c** SBP of archaeal *Pfu*SAHH in complex with SIH (PDB ID: 7R38). **d** SBP of eukaryotic *Mm*SAHH in complex with inosine (INO). $Na^+$ is shown as blue sphere (PDB ID: 8COD). **e** In *Pfu*SAHH, the cofactors are loosely covered by the adjacent monomer while some room is visible between the monomers, indicating a less stable enzyme (PDB ID: 7R38). **f** In *Mm*SAHH, the cofactors are tightly covered by the C-terminus of the adjacent monomer (PDB ID: 5AXA[53]).

between the monomers of the archaeal SAHHs show a cleft, when looking at the surface representation (Fig. 7e). This is not visible in eukaryotic SAHHs, such as the one from mouse (Fig. 7f); also because of the longer C-terminus that covers the cofactor. This means the cofactor is more exposed to the environment in archaea, and indicates a less stable tetrameric form for archaeal SAHHs/SIHHs, as fewer interactions are found between the monomers. *Tm*SAHH was found to require a much higher concentration of NAD⁺, which was linked back to the shortened C-terminus, impacting the binding affinity[27]. *Tm*SAHH has the shortest C-terminus, which is three amino acids shorter than the archaeal ones. All SAHHs/SIHHs in this work (including *Tm*SAHH) were active without the addition of NAD⁺ at 37 °C and thus no dependency between concentration of NAD⁺ and length of C-terminus was observed in our hands. Moreover, recombinant *Tm*SAHH was described to only be enzymatically active after a heat-induced conformational change[27,42]. To further investigate this observation, we used recombinant enzymes either previously preincubated at 95 °C or permanently handled at room temperature for crystallisation of the hyperthermophilic *Pfu*SAHH in complex with inosine. Comparison of both structures showed no substantial differences in the conformation of the differently treated enzymes (Supplementary Fig. S30 online). Independent of the previous thermal treatment, *Pfu*SAHH, as the other archaeal homologues, formed stable homotetramers in the crystal. To verify the arrangement of individual units in the multimeric assembly of the archaeal SAHH crystal structures, tetramers were constructed and analysed for each SAHH individually in their respective space groups using PDBePISA[58]. The

prediction analysis of *Pfu*SAHH tetramers (both inosine and SIH complexes) showed an average value for the solvation free energy on formation of the assembly ($\Delta G^{int}$) of −129.0 kcal mol⁻¹. Similarly, *Mma*SAHH tetramers and *Sac*SAHH tetramers exhibited an average $\Delta G^{int}$ value of −133.5 kcal mol⁻¹ and −127.5 kcal mol⁻¹, respectively. In comparison, selected eukaryotic SAHHs with a long C-terminus, such as *Mm*SAHH tetramers (both inosine and adenosine complexes), showed an average $\Delta G^{int}$ of −186.7 kcal mol⁻¹. These predictions suggest that the eukaryotic SAHH tetramers display more favourable solvation free energy than the archaeal SAHH tetramers. Regarding the stability of archaeal SAHH tetramers, the shortened C-terminal domain was speculated to either be useful for intramolecular motion at high temperature and prevent denaturation based on a protein model for *Pfu*SAHH[36], or to positively affect the thermostability of the enzyme[27]. This is in accordance with the analyses provided in this study.

## Conclusion

In this work, we present the biochemical characterisation of various archaeal, bacterial, and eukaryotic SAHH/SIHH homologues revealing different substrate preferences. While archaeal enzymes deriving from the Crenarchaeota phylum (*Sac*SAHH and *Sso*SAHH) show catalytic activity only with SAH, homologues from *T. maritima* and the euryarchaeal classes of *Methanococci* and *Thermococci* (*Mi*SAHH, *Mj*SIHH, *Pfu*SAHH, *Tk*SAHH, and *Mma*SAHH) show higher catalytic activity with SIH. Our biochemical characterisation together with the bioinformatical analyses indicate that multiple more euryarchaeal organisms employ a pathway for

methyl metabolism and purine salvage using a deamination step. As some archaea evolutionary prefer purine salvage pathways with inosine and hypoxanthine as intermediates, this might not only apply for the metabolism of SAM but various other cofactors. Moreover, it is reasonable to investigate the incorporation of hypoxanthine-derivatives into other enzyme cofactors in these organisms. Similar pathways have been actually observed in the archaeon *Thermococcus kodakarensis*, in which a deamination of NAD$^+$ has been reported[59]. In addition to the biochemical investigations, we provide structural insights into SAHHs/SIHHs originating from archaeal organisms. The archaeal enzymes show the same binding modes for the cofactor NAD$^+$ as well as for the substrates inosine/adenosine and SIH/SAH as homologues from other domains of life. The absence of a monovalent cation in proximity to the active site suggests that the catalytic activity of archaeal SAHHs/SIHHs is independent of monovalent cations. The shortened C-terminal domain of archaeal SAHHs/SIHHs has an influence on their tertiary and quaternary arrangement showing differences compared to bacterial and eukaryotic homologues. In summary, our functional and structural knowledge of SAHHs/SIHHs strongly promotes the existence of an alternate route for methyl metabolism and purine salvage, which might run in parallel with the canonical pathway, as SAH is also accepted as a substrate.

## Methods
### Materials
Substrates and reference standards were purchased in the highest purity available from Sigma-Aldrich (adenosine, Hcy, hypoxanthine, inosine, SAH, *S*-methyl-L-methionine), and AppliChem (adenine). Ingredients for buffers and cultivation media were purchased from Carl Roth.

### Cloning, expression, and protein purification
Genes of *Cg*SAHH, *Ll*SAHH, *Mj*SIHH, *Mma*SAHH, *Mm*SAHH, *Pa*SAHH, *Pfu*SAHH, *Sac*SAHH, *Sso*SAHH, *Tm*SAHH, and *Mj*DadD were purchased as synthetic DNA strings from Invitrogen (Thermo Fisher Scientific, Waltham, MA, USA). *Sa*SAHH, *Sf*SAHH and *Tk*SAHH were cloned directly from genomic DNA. The genes were amplified by PCR and cloned into pET28a(+) using T4 DNA ligase (New England Biolabs GmbH, Frankfurt am Main, Germany) or In-Fusion Cloning (Takara Bio Europe, Saint-Germain-en-Laye, France). All primers are listed in Supplementary Table S1 online. For *Mc*SAHH, *Me*SAHH, *Mh*SAHH, *Mi*SAHH, and *Mt*SAHH, pET28a(+) based plasmids encoding the enzymes were ordered from BioCat GmbH (Heidelberg, Germany). The L-homocysteine *S*-methyltransferase from *Saccharomyces cerevisiae* (*Sc*HSMT) was cloned from genomic DNA of baker's yeast (primers used: *Sc*HSMT-NdeI-fwd 5'-T ATATACATATGAAGCGCATTCCAATCAAAG-3' and *Sc*HSMT-HindIII-rev 5'-TATATAAGCTTAGGAGTATTTATCTACAGCTGAT GC-3') into pET28a(+) using T4 DNA ligase (New England Biolabs GmbH, Frankfurt am Main, Germany). The enzymes were produced in *E. coli* BL21-Gold(DE3) competent cells (Agilent, Santa Clara, CA, USA). The expressions of *Sa*SAHH, *Sf*SAHH and *Tk*SAHH were performed in *E. coli* BL21-CodonPlus (DE3)-RIPL competent cells (Agilent, Santa Clara, CA, USA). LB medium was used for most seed cultures and main cultures, *Sa*SAHH, *Sf*SAHH and *Tk*SAHH were produced in 2xYT medium. Seed cultures (5 mL) were grown in LB medium with the corresponding antibiotics at 37 °C overnight. The main culture (400 mL plus added seed culture) was grown in medium supplemented with the needed antibiotics at 37 °C. When the OD$_{600}$ reached 0.5, expression was induced by the addition of isopropyl-β-D-thiogalactopyranoside (IPTG; final concentration 0.25 mM, 1 mM for *Pfu*SAHH) and the cultures were shaken at 160 rpm for 18 h at 20 °C. The cells were harvested by centrifugation and lysed by sonication [Branson Sonifier 250, Emerson, St. Louis, MO, USA (duty cycle 50%, intensity 50%, 5 × 30 s with 30 s breaks)] in lysis buffer. The lysis buffer was either 50 mM Tris-HCl, pH 7.4, 500 mM NaCl, 10% (*w/v*) glycerol for *Mc*SAHH, *Me*SAHH, *Mh*SAHH, *Mi*SAHH, *Mj*SIHH, *Mma*SAHH, *Mt*SAHH, *Pfu*SAHH, *Sac*SAHH, *Sso*SAHH and *Tk*SAHH; or 40 mM Tris-HCl, pH 8.0, 100 mM NaCl, 10% (*w/v*) glycerol for *Cg*SAHH, *Ll*SAHH, *Mj*DadD, *Mm*SAHH, *Pa*SAHH, *Sa*SAHH, *Sc*HSMT, *Sf*SAHH, and *Tm*SAHH. After

centrifugation, the proteins were purified by nickel-NTA affinity chromatography [lysis buffer including low concentrations (10 or 10–50 mM) imidazole for washing or high concentrations (200 or 100–300 mM) for eluting the protein], and desalted using a PD-10 column (Cytiva Europe GmbH, Freiburg im Breisgau, Germany). The storage buffer for the enzymes was the same as the lysis buffer. Protein concentration was determined using a NanoDrop 2000, at 280 nm with the molecular weight (including the His$_6$-tag, Supplementary Table S1 online), and the protein extinction coefficient (calculated with the ExPASy ProtParam tool[60]).

### Crystallisation and data collection
*Pfu*SAHH with inosine (2 mM) co-crystals appeared after 2–4 days by using the sitting drop vapor diffusion methods at room temperature by combining 0.5 µL of protein at 10 mg/mL with 0.5 µL of a precipitant solution comprising 26% (*w/v*) PEG 1500 with 100 mM malic acid/MES/Tris-HCl buffer (MMT), pH 8.0. *Pfu*SAHH treated at 95 °C (368 K) for 15 min was also co-crystallized with inosine and the crystals appeared after 2–4 days. *Pfu*SAHH with SIH co-crystals were obtained with slightly modified crystallisation conditions comprising 28% (*w/v*) PEG 2000 with 100 mM MMT, pH 8.0. *Mma*SAHH (10 mg/mL) with inosine (2 mM) co-crystals were obtained within a week at 22 °C in sitting drops by mixing 0.5 µL of the protein solution with an equal volume of reservoir solution containing 28% (*w/v*) PEG 3350, 100 mM MMT, pH 8.0. *Sac*SAHH (10 mg/mL) with adenosine (2 mM) co-crystals were obtained within a week at 22 °C in sitting drops by mixing 0.5 µL of the protein solution with an equal volume of reservoir solution containing 20% (*w/v*) PEG 3350 with 200 mM ammonium iodide. *Mm*SAHH with inosine (4 mM) co-crystals appeared after 4 days by using the sitting drop vapor diffusion methods at room temperature by combining 2 µL of protein at 4 mg/mL with 1 µL of a precipitant solution comprising 22% (*w/v*) PEG 3350 with 180 mM sodium formate, pH 6.9. Prior to data collection, the crystals were transferred to a cryosolution containing the respective mother liquor reservoir solutions and flash frozen in liquid nitrogen. Datasets were collected at 100 K at the Swiss Light Source (SLS) on macromolecular crystallography beamline PXI-X06SA. All *Pfu*SAHH crystals were maintained at a constant temperature (100 K) and a total of 900 images ($\Delta\varphi = 0.2°$/image) for inosine complex, and 1800 images ($\Delta\varphi = 0.2°$/image) for the SIH complex were recorded separately for each on an EIGER 16 M (Dectris) detector. The datasets were extending up to 2.3 Å resolution for inosine complex, and up to 2.0 Å for the SIH complex. All datasets were processed by XDS[59] in $P4_22_12$ space group (a = b = 111.68, c = 121.59; α = β = γ = 90°). *Mma*SAHH crystals were flash-cooled and maintained at a constant temperature at 100 K in a cold nitrogen-gas stream. A dataset with a total of 900 images ($\Delta\varphi = 0.2°$/image) were recorded on an EIGER 16 M (Dectris) detector. The datasets were extending up to 2.5 Å resolution in $P2_1$ space group (a = 65.76, b = 328.9, c = 82.05; α = γ = 90°, β = 107.2°). *Sac*SAHH crystals were maintained at a constant temperature (100 K) and a total of 3600 images ($\Delta\varphi = 0.1°$/image) were recorded on a EIGER 16 M (Dectris) detector with data extending up to 2.6 Å resolution. The datasets were processed in $P1$ space group (a = 84.36, b = 88.36, c = 138.54; α = 78.86°, β = 74.85°, γ = 64.85°). *Mm*SAHH crystals were maintained at a constant temperature (100 K) and a total of 900 images ($\Delta\varphi = 0.1° \cdot$image$^{-1}$) were recorded on an EIGER 16 M (Dectris) detector with data extending up to 2.48 Å resolution. The datasets were processed by XDS[59] in $I222$ space group (a = 97.78, b = 101.96, c = 172.58; α = β = γ = 90°). All the data were integrated by using XDS[61] then merged and scaled using SCALA from the CCP4 suite of programs[62,63]. The data collection statistics are summarised in Table 3.

### Structure determination and refinement
Initial phases were determined by molecular replacement using PHASER. Best solutions were obtained using 5AXA based homology model as the starting model for *Pfu*SAHH, 1V8B for *Mma*SAHH and 3H9U for *Sac*SAHH. The asymmetric unit (ASU) of *Pfu*SAHH crystals contained a dimer, whereas *Mma*SAHH comprised four dimers and SacSAHH comprised two dimers. The models were built with COOT[63], and refinements were carried out

**Table 3 | Data collection and refinement statistics (Molecular Replacement)**

| | MmaSAHH•NAD•inosine | PfuSAHH•NAD•inosine | PfuSAHH•NAD•inosine (95 °C) | PfuSAHH•NAD•SIH | SacSAHH•NAD•adenosine | MmSAHH•NAD•inosine |
|---|---|---|---|---|---|---|
| **Data collection** | | | | | | |
| Space group | $P2_1$ | $P4_22_12$ | $P4_22_12$ | $P4_22_12$ | $P1$ | $I222$ |
| Cell dimensions | | | | | | |
| a, b, c (Å) | 65.8, 328.9, 85.1 | 112.5, 112.5, 122.8 | 111.7, 111.7, 122.1 | 111.7, 111.7, 121.6 | 84.4, 88.4, 138.6 | 98.2, 102.5, 173.4 |
| $\alpha$, $\beta$, $\gamma$ (°) | 90, 107.2, 90 | 90, 90, 90 | 90, 90, 90 | 90, 90, 90 | 78.9, 74.9, 64.9 | 90, 90, 90 |
| Resolution [Å] | 45.44–2.65 (2.74–2.65) | 48.62–2.28 (2.37–2.28) | 48.31–2.03 (2.11–2.03) | 53.4–2.05 (2.12–2.05) | 46.16–2.50 (2.59–2.50) | 54.9–2.48 (2.56–2.48) |
| $R_{merge}$ | 0.1748 (1.075) | 0.1495 (1.191) | 0.2373 (2.425) | 0.1048 (0.9204) | 0.1348 (0.582) | 0.177 (0.97) |
| $I/\sigma I$ | 6.73 (2.13) | 13.51 (1.66) | 15.36 (1.54) | 16.82 (2.70) | 6.67 (2.35) | 7.8 (2.1) |
| Completeness [%] | 97.3 (98.2) | 99.8 (98.8) | 99.7 (97.6) | 99.7 (99.9) | 87.5 (85.9) | 98.8 (99) |
| Redundancy | 3.6 (3.5) | 12.7 (8.3) | 26.8 (26.6) | 12.8 (13.4) | 3.6 (3.6) | 5.7 (5.9) |
| **Refinement** | | | | | | |
| Resolution [Å] | 45.44–2.65 (2.74–2.65) | 48.62–2.28 (2.37–2.28) | 48.31–2.03 (2.11–2.03) | 53.4–2.05 (2.12–2.05) | 46.16–2.50 (2.59–2.50) | 54.9–2.48 (2.56–2.48) |
| No. reflections | 141280 (13525) | 36333 (3519) | 50113 (4818) | 48676 (4780) | 105739 (10341) | 31048 (3056) |
| $R_{work}/R_{free}$ [%] | 18.8/24.3 | 16.1/21.6 | 15.5/20.3 | 17.6/21.9 | 22.9/28.2 | 17.0/21.8 |
| No. atoms | 26322 | 6891 | 7086 | 7147 | 27556 | 7067 |
| Protein | 25440 | 6604 | 6660 | 6706 | 25760 | 6650 |
| Ligand/ion | 608 | 126 | 126 | 88 | 504 | 128 |
| Water | 274 | 161 | 300 | 353 | 1292 | 289 |
| B-factors | 67.14 | 46.77 | 35.81 | 38.41 | 40.96 | 40.98 |
| Protein | 67.59 | 46.98 | 35.67 | 38.16 | 40.23 | 41.16 |
| Ligand/ion | 55.84 | 38.49 | 28.84 | 34.81 | 31.83 | 35.81 |
| Water | 50.24 | 44.84 | 41.95 | 44.11 | 59.02 | 39.23 |
| R.m.s. deviations | | | | | | |
| Bond lengths (Å) | 0.030 | 0.014 | 0.009 | 0.014 | 0.008 | 0.014 |
| Bond angles (°) | 2.33 | 1.81 | 1.48 | 1.84 | 1.69 | 1.86 |

Each structure was solved from a single crystal. Values in parentheses are for highest-resolution shell.

with REFMAC using NCS constraints with or without TLS parameters[64]. The NAD$^+$ ligands in all the complex crystals were clearly observed in the initial 2Fo-Fc and Fo-Fc maps. To improve the model of the bound substrate and product, we calculated both 2Fo-Fc maps and POLDER omit maps[65] and fitted the ligands into the respective electron densities, which allowed unambiguous identification of the ligand positioning (Supplementary Fig. S31 online). Incorporation of non-crystallographic symmetry (NCS) restraints greatly expedited model improvement. For *Pfu*SAHH•NAD•inosine complex, a Ramachandran plot calculation indicated that 97% and 3% of the residues occupy the most favored and additionally allowed regions, respectively. For *Pfu*SAHH•NAD•SIH complex, a Ramachandran plot calculation indicated that 96% and 4% of the residues occupy the most favored and additionally allowed regions, respectively. For *Mma*SAHH•NAD•inosine complex, a Ramachandran plot calculation indicated that 96% and 3% of the residues occupy the most favored and additionally allowed regions, respectively. For *Sac*SAHH•NAD•adenosine complex, a Ramachandran plot calculation indicated that 94% and 5% of the residues occupy the most favored and additionally allowed regions, respectively. For *Mm*SAHH•NAD•inosine complex, a Ramachandran plot calculation indicated that 97% and 3% of the residues occupy the most favored and additionally allowed regions, respectively. Analysis of the SAHH structures and comparison with other SAHH structures were carried out using PyMOL[66].

### NMR spectroscopy
Nuclear magnetic resonance (NMR) spectra were recorded on an Avance DRX 400 spectrometer (Bruker, Billerica, MA, USA), operating at 400.1 MHz (for $^1$H NMR) and 100.6 MHz (for $^{13}$C NMR). All measurements were performed at 25 °C. Spectra were analysed with TopSpin 3.6.2.

### HPLC analysis
All available substrates and products of the enzymatic reactions are used as authentic reference standards and the retention times are listed in Supplementary Table S3 online. All assays were analysed with an Agilent 1100 Series HPLC using an ISAspher SCX 100-5 column (250 mm × 4.6 mm, 5 µm; ISERA GmbH, Düren, Germany). For the HPLC method[67], 40 mM sodium acetate, pH 4.2, (buffer A), and acetonitrile (buffer B) were used as mobile phase. A stepwise gradient [0–4 min: 100% A (1.3 mL/min); 4–10 min:70% A, 30% B, (1.1 mL/min); 10–20 min: 100% A (1.3 mL/min, re-equilibration)] was used for the elution. The injection volume was set to 10 µL.

### SIH enzymatic synthesis and structure verification
SIH was enzymatically synthesised with a 5′-deoxyadenosine deaminase (*Mj*DadD) starting from SAH following a published protocol with modifications[13,14]. 1 mM SAH was incubated with 1 µM *Mj*DadD in 50 mM Tris-HCl buffer, pH 8.0, for 20 h at 37 °C. The enzyme was removed with a spin filter and full conversion checked with HPLC-DAD. The stock solution of SIH (1 mM) was stored at −20 °C. The structure was verified by NMR analysis by running six parallel assays with 4 mL reaction tube and combining them in one 100 mL flask. The solution was freeze-dried, and the powder was resuspended in 700 µL D$_2$O and measured with NMR spectroscopy. SIH has been isolated from *Streptomyces flocculus* (*Streptomyces albus* ATCC 13257) previously and the structure was confirmed by UV spectrum and $^1$H- NMR analysis[16]. An HPLC method was used to track the conversion from SAH to SIH catalysed by *Mj*DadD (Supplementary Fig. S1 online), here the UV spectra matched the described absorbance maximum shift from 260 nm (SAH) to 248 nm (SIH; Supplementary Fig. S1 online). $^1$H NMR data (Supplementary Fig. S2B online) obtained matched the published data and was extended by measuring the $^{13}$C-NMR spectrum (Supplementary Fig. S2C online), as well as 2D spectra (Supplementary Figs. S2D–F online) to further confirm the SIH structure.

### Enzyme activity assays
In general, all assays were performed at least in triplicates. Assays concerning SAHHs/SIHHs were performed in 50 mM Tris-HCl, pH 7.5, for 20 h at 37 °C in 200 µL reaction volume, if not otherwise stated. For the synthesis reaction, 0.5 mM Hcy and 0.5 mM nucleoside were added, while the cleavage reaction was started with 0.2 mM of either SAH or SIH. The SAHH/SIHH was added at 5 µM. A second enzyme, *Sc*HSMT (10 µM), and its methyl donor *S*-Methyl-L-methionine (1 mM) were added to the SAH or SIH cleavage to drive the reaction forward. For assays performed at 70 °C, *Sc*HSMT was replaced by DTNB (300 µM). After incubation, 150 µL of the assay sample was added to 50 µL of perchloric acid [final concentration 2.5% (w/v)] to stop the reaction and spun down for 30 min prior transferring 80 µL to an HPLC vial. All investigated SAHHs and SIHHs were tested in the SAH and SIH cleavage, as well as synthesis direction. An overview can be found in Table 1, while the HPLC chromatograms are given in Supplementary Figs. S4–S21 and the SDS gels in Supplementary Fig. S3 online.

### Bioinformatical analysis of amino acid sequences
Alignments of amino acid sequences were performed either with Clustal Omega or T-Coffee online services[68,69]. To visualise and annotate the sequence alignment, Jalview Version2 was used[70]. For the visualisation of the phylogenetic tree, the MEGA11 software was used[71].

### Statistics and reproducibility
All the experiments were performed at least in triplicates. Substrate conversions were analysed semi-quantitative by using peak areas. All data presented here were reliably reproducible.

### Reporting summary
Further information on research design is available in the Nature Portfolio Reporting Summary linked to this article.

### Data availability
The single chromatograms used to determine the substrate preferences can be found in the Supplementary information online (Fig. S4: *Sac*SAHH, Fig. S5: *Sso*SAHH, Fig. S6: *Mc*SAHH, Fig. S7: *Me*SAHH, Fig. S8: *Mh*SAHH, Fig. S9: *Mi*SAHH, Fig. S10: *Mj*SIHH, Fig. S11: *Mma*SAHH, Fig. S12: *Mt*SAHH, Fig. S13: *Pfu*SAHH, Fig. S14: *Tk*SAHH, Fig. S15: *Cg*SAHH, Fig. S16: *Pa*SAHH, Fig. S17: *Sa*SAHH, Fig. S18: *Sf*SAHH, Fig. S19: *Tm*SAHH, Fig. S20: *Ll*SAHH, Fig. S21: *Mm*SAHH). Source data underlying the chromatograms presented in Fig. 3b and c as well as the Supplementary Figs. S4–S24 can be found in Supplementary Data 1 or online in the data Repository of the University of Stuttgart[72]. Supplementary Figs. S33 and S34 contain the original uncropped and unedited gel images of Supplementary Fig. S3. Maps and models have been deposited in the PDB with the accession codes: 7R37, 7R38, 7R39,7R3A, 8COD and 8QNO. All other data supporting the findings of this study are available from the corresponding author upon reasonable request.

### Code availability
Protein homologues were searched using NCBI BLASTP[47] with the standard algorithm parameters. To obtain more relevant results, the number of 'Max target sequences' and the 'Expect threshold' were set to 1,000 and 0.01, respectively. Alignments of amino acid sequences were performed either with Clustal Omega or T-Coffee online services[68,69] using the standard parameters. To visualise and annotate the sequence alignment, Jalview Version2 was used[70]. For the visualisation of the phylogenetic tree, the MEGA11 software was used[71]. Analysis of the SAHH structures and comparison with other SAHH structures were carried out using PyMOL[66].

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

## Acknowledgements

This work was supported by the Deutsche Forschungsgemeinschaft (DFG,
German Research Foundation) – 235777276/RTG1976 and 527572100.
This project has received funding from the European Research Council
(ERC) under the European Union's Horizon 2020 research and innovation
programme (Grant agreement No. 716966). We thank Jun.-Prof. Dr Silja
Mordhorst (now University of Tübingen) for construction of some plasmids
used in this work, and Prof. Dr Andreas Bechthold for the donation of gDNA
from *S. albus* J1074. We thank Emina Čokljat and Katharina Strack for
assistance with protein production and purification. We thank Dr. Tomizaki
Takashi and the PXI (X06SA) beamline staff of the Swiss Light Source, Paul
Scherrer Institute (Villigen, Switzerland) for support in crystallographic data
collection. We thank Sascha Ferlaino and Dr Philippe Bisel (both University
of Freiburg) for NMR measurements and interpretation of the results,
respectively. Prof. Dr Sonja-Verena Albers (University of Freiburg) is
acknowledged for sharing her expertise regarding archaea and for critically
reading the manuscript; as well as Prof. Dr Michael Müller (University of
Freiburg) for helpful discussions on the binding modes and mechanisms.

## Author contributions

L.-H.K.: conceptualisation, methodology, investigation; writing - review and
editing, and visualisation; D.P.: conceptualisation, methodology,
investigation, writing - original draft preparation, writing - review and editing,
and visualisation; R.S.-B.: conceptualisation, methodology, investigation,
writing - original draft preparation, writing - review and editing, and
visualisation; P.G.: data interpretation, and writing - review and editing;
J.N.A.: conceptualisation, writing - review and editing, supervision, project
administration, resources, and funding acquisition.

## Funding

## Competing interests

The authors declare no competing interests.

## Additional information

**Supplementary information** The online version contains
supplementary material available at

Jennifer N. Andexer.

**Peer review information** *Communications Biology* thanks Jeremy Lott,
Krzysztof Brzeziński, and Kozo Tomita for their contribution to the peer
review of this work. Primary Handling Editors: Isabelle Lucet and Tobias
Goris. A peer review file is available.

**Publisher's note** Springer Nature remains neutral with regard to
jurisdictional claims in published maps and institutional affiliations.

