## [Peer review file · Communications Biology]

Reviewers' comments:

Reviewer #1 (Remarks to the Author):

SAHase is involved in the regulation of cellular SAM-dependent methylation. Structural studies of SAHases of various origins have been conducted for over 20 years. However, no experimental model of any archaeal enzyme has been elucidated so far. This paper fills these gaps and describes X-ray crystallographic studies of several SAHases from the Euryarcheota and Crenarcheota phyla with enzyme activity assays for various SAHase homologs derived from archaeal, bacterial and eukaryotic organisms. The authors revealed differences in substrate preferences between SAHases of various origins. Notably, the authors indicated that S-inosine-L-homocysteine (SIH) is a better substrate than SAH for the enzymes from Euryarcheota and *Thermotoga maritima* (phylogeny related). On the other hand, similarly to bacterial and eukaryotic enzymes, SAH is a preferred substrate for SAHases from Crenarcheota. These observations allowed the authors to obtain crystal structures of SAHases under investigation in a complex with adenosine, inosine or SIH. These findings support other studies on alternative SAH metabolism pathways and regulation of archaea cellular methylation processes. The paper is generally well-written, biochemical and structural studies are technically sound, and the results are of general interest. The paper is a strong candidate for this journal if the issues below are adequately addressed in a revised version.

1. Previous structural studies related to hyperthermophilic SAHase from *Thermotoga maritima* (Zheng et al., 2015, doi: 10.1016/j.jsb.2015.03.002 (ligand-free TmSAHase) and Brzezinski et al., 2017, doi: 10.1016/j.ijbiomac.2017.06.065 (adenosine-bound TmSAHase)) indicated that the enzyme expressed and purified at room temperature (that is non-physiological for the bacterium) requires thermal activation (heat-induced conformational changes are required to attain enzymatic activity). Therefore, I have some questions regarding enzymatic activity assays:

1.1. The authors performed all tests at 37 degrees C. Why did the authors not perform those assays at physiological temperatures, especially for the enzymes derived from thermophilic organisms?

1.2. Following the previous question, what do the authors think about the relationship between substrate specificity and temperature of the reaction environment (also, in light of heat-induced conformational changes of the enzyme)? Can SAH or SIH hydrolysis/synthesis rates be different at various temperatures?

2. SAH concentration needs to be controlled to ensure proper cellular methylation. This control function is fulfilled by methylthioadenosine/SAH nucleosidase (MTAN) or/and SAHase. Depending on the organism, the genomes encode both enzymes or only one of them. It could be convenient to analyze if archaeal genomes carry genes encoding MTAN. It could hint if SAHase homologs from archaea, especially those from Euryarcheota, have a single (SIH metabolism) or dual (SIH and SAH metabolism) function.

3. Most SAHases of bacterial and eukaryotic origin require a monovalent cation (preferably potassium) for catalytic activity. Does the same apply to archaeal enzymes? Could the authors give some comments on that?

4. Are there any significant geometrical differences between the macromolecular environment of the substrate binding pocket of MmSAHH complexed with adenosine and inosine?

5. A figure of superposed substrate binding pockets of SAHases of bacterial, eukaryotic and archaeal SAHases would be helpful to illustrate significant differences in ligand binding.

Reviewer #2 (Remarks to the Author):

Popadic et al. deals with the structural and biochemical characteristics of S-adenosyl-L homocysteine hydrolases (SAHHs) and S-inosyl-L homocysteine hydrolases (SIHHs). These enzymes play a crucial role in the regulation of the methyl cycle within cells by hydrolyzing SAH (or SIH) into adenosine (or inosine) and L-homocysteine.

The manuscript presents crystal structures of SAHHs from archaeal sources and their complexes with cofactors. Comparative analysis of these structures with bacterial and eukaryotic SAHHs is also conducted.

This reviewer appreciates the paper's meticulous execution and technically sound structural and biochemical analyses. Having said that, this reviewer feels that this manuscript falls short in providing us of the understanding of the biological significance of these enzyme families. This reviewer also feels that the manuscript does not advance our understanding of this enzyme in the cellular metabolism of SAM, which governs various cellular phenomena.

This reviewer suggests submitting the manuscript to a more specialized journal, as *Communication Biology* may not be the most suitable venue for this study.

Reviewer #3 (Remarks to the Author):

The introduction to the paper is exceptionally well written and was a pleasure to read. However, the main body of the paper requires editing to focus more clearly on the data being presented and to strip out excess detail. At some points the text felt like a combination of research paper and literature review, and it was difficult to tell which data were new and which were from the literature.

The main data presented are the crystal structures of three archaeal SAH hydrolase enzymes, plus the structure of the mouse SAH hydrolase in complex with inosine. Additionally, semi-quantitative enzyme assays were carried out on a large range of eukaryotic, bacterial and archaeal enzymes. However, there is an incongruity between the presented sequence-based phylogenetic tree and the observed enzyme phenotypes, expressed here in terms of substrate preference, that is not satisfactorily resolved. The structures of the archaeal enzymes are described in great detail, as are the details of substrate binding. This information could have been much more succinctly described with a greater and more information-dense use of diagrams and data-summarising tables. Overall, despite the large volume of carefully conducted work, the conclusions drawn seem rather underwhelming, and unfortunately are not likely to be of strong interest to those in the wider scientific community.

21.12.2023

Point-to-point reply (manuscript ID: COMMSBIO-23-1804)

→ Our replies to comments in BLUE

Reviewer 1

SAHase is involved in the regulation of cellular SAM-dependent methylation. Structural studies of SAHases of various origins have been conducted for over 20 years. However, no experimental model of any archaeal enzyme has been elucidated so far. This paper fills these gaps and describes X-ray crystallographic studies of several SAHases from the Euryarcheota and Crenarcheota phyla with enzyme activity assays for various SAHase homologs derived from archaeal, bacterial and eukaryotic organisms. The authors revealed differences in substrate preferences between SAHases of various origins. Notably, the authors indicated that S-inosine-L-homocysteine (SIH) is a better substrate than SAH for the enzymes from Euryarcheota and *Thermotoga maritima* (phylogeny related). On the other hand, similarly to bacterial and eukaryotic enzymes, SAH is a preferred substrate for SAHases from Crenarcheota. These observations allowed the authors to obtain crystal structures of SAHases under investigation in a complex with adenosine, inosine or SIH. These findings support other studies on alternative SAH metabolism pathways and regulation of archaea cellular methylation processes. The paper is generally well-written, biochemical and structural studies are technically sound, and the results are of general interest. The paper is a strong candidate for this journal if the issues below are adequately addressed in a revised version.

1. Previous structural studies related to hyperthermophilic SAHase from *Thermotoga maritima* (Zheng et al., 2015, doi: 10.1016/j.jsb.2015.03.002 (ligand-free TmSAHase) and Brzezinski et al., 2017, doi: 10.1016/j.ijbiomac.2017.06.065 (adenosine-bound TmSAHase)) indicated that the enzyme expressed and purified at room temperature (that is non-physiological for the bacterium) requires thermal activation (heat-induced conformational changes are required to attain enzymatic activity). Therefore, I have some questions regarding enzymatic activity assays:

1.1. The authors performed all tests at 37 degrees C. Why did the authors not perform those assays at physiological temperatures, especially for the enzymes derived from thermophilic organisms?

→ Originally, we chose 37 °C as a temperature that was (in our experience from in vitro multienzyme systems) suitable for most enzymes. We agree that for thermophilic organisms, it might be interesting to extend this, also in terms

of the enzymes' potential applicability of biocatalysts at elevated temperature. We have now repeated the assays for two selected enzymes from the thermophilic organisms at 70 °C and incorporated the results accordingly (see SI page S32 and S33, Figure S23A, S23B, S24A and S24B). In this case, increased temperatures do not influence the substrate preferences (see MS page 8 (line 23-42) and page 9 (line 1-3) starting with "Increased temperatures do not influence the substrate preferences, line 23").

Figure S23A. HPLC chromatograms showing A. the SAH synthesis reaction and B. the SAH cleavage reaction (with and without the addition of DTNB) catalysed by PfuSAHH at 70 °C.

Figure S23B. HPLC chromatograms showing A. the SIH synthesis reaction and B. the SIH cleavage reaction (with and without the addition of DTNB) catalysed by PfuSAHH at 70 °C.

Figure S24A. HPLC chromatograms showing A. the SAH synthesis reaction and B. the SAH cleavage reaction (with and without the addition of DTNB) catalysed by SacSAHH at 70 °C.

Figure S24B. HPLC chromatograms showing A. the SIH synthesis reaction and B. the SIH cleavage reaction (with and without the addition of DTNB) catalysed by SacSAHH at 70 °C.

1.2. Following the previous question, what do the authors think about the relationship between substrate specificity and temperature of the reaction environment (also, in light of heat-induced conformational changes of the enzyme)? Can SAH or SIH hydrolysis/synthesis rates be different at various temperatures?

→ The only difference we observed between the two temperatures tested was a low conversion of SIH by the enzyme from *Sulfolobus acidocaldarius*, which was not detected at 37 °C. This could be due to the higher activity of the enzymes at elevated temperature, assuming that it was under the detection limit in the 37 °C sample; however, the activity was rather low and far from a preferred conversion of SIH/inosine. To double-check on potential heat-induced conformational changes of thermophilic SAHs, we added data on a crystal structure of *PfuSAHH* treated at 95 °C before crystallisation. The comparison of the two structures did not show any substantial differences (see MS page 14 (Line 11-17) starting with “Moreover, recombinant *TmSAHH* was described to only be enzymatically active after a heat-induced conformational change, (Line 11)” and SI page S45, Figure S30). Moreover, all (hyper-)thermophilic enzymes (curiously including the enzyme from *Thermotoga maritima*) tested in our study were catalytically active without previous heat-activation.

Figure S30. Crystal structures of PfuSAHH treated at different temperatures. A. Tetrameric assembly of PfuSAHH treated at 95 °C (PDB ID: 8QNO); B. Tetrameric assembly of PfuSAHH treated at room temperature (PDB ID: 7R37). C. Monomer of PfuSAHH treated at 95 °C in the closed conformation. D. Monomer of PfuSAHH treated at room temperature in the closed conformation.

2. SAH concentration needs to be controlled to ensure proper cellular methylation. This control function is fulfilled by methylthioadenosine/SAH nucleosidase (MTAN) or/and SAHase. Depending on the organism, the genomes encode both enzymes or only one of them. It could be convenient to analyze if archaeal genomes carry genes encoding MTAN. It could hint if SAHase homologs from archaea, especially those from Euryarcheota, have a single (SIH metabolism) or dual (SIH and SAH metabolism) function.

→ We took this advice and analysed whether MTAN is encoded in archaeal genomes. A NCBI BLASTP search for archaeal homologues of MTAN from *Escherichia coli* revealed 32 archaea encoding an enzyme annotated as MTAN. Out of these organisms, six belong to the phylum of Euryarchaeota. None of the SAHs from those organisms were part of our research. Taken together, these results strongly indicate that (eury)archaeal SAHH homologues have a dual function by metabolising SAH as well as SIH. We have included and discussed the relevant results of the BLASTP search in the revised manuscript (see MS page 9 (Line 13-26) starting with "...This is inconsistent with the substrate scope of characterised archaeal SAHs/SIHHs as the subgroup of tested euryarchaeal McSAHH, MeSAHH, MhSAHH and MtSAHH showed no activity for SIH cleavage" and SI page S37 and S38, Table S7).

Table S7 Homologues of EcMTAN found in the genomes whose SAHH/SIHH was characterised in this work, using BLASTP³⁶. Sequence identity and similarity were calculated using the Emboss Needle algorithm⁷.

Organism	UniProt accession number	Sequence identity/similarity [%] to EcMTAN
Corynebacterium glutamicum	A0A1R4F352_CORGT	21.9/36.8
Lupinus luteus	B6DX57_LUPLU	23.0/40.9
Methanocaldococcus jannaschii	Not available	Not available
Methanococcus maripaludis	Not available	Not available
Mus musculus	Not available	Not available
Methanocella conradii	Not available	Not available
Methanohalobium evestigatum	Not available	Not available
Methanohalophilus halophilus	Not available	Not available
Methanocaldococcus infernus	Not available	Not available
Methanotherix thermoacetophila	Not available	Not available
Pseudomonas aeruginosa	Not available	Not available
Pyrococcus furiosus	Not available	Not available
Sulfolobus acidocaldarius	Not available	Not available

Organism	UniProt accession number	Sequence identity/similarity [%] to EcMTAN
Corynebacterium glutamicum	A0A1R4F352_CORGT	21.9/36.8
Lupinus luteus	B6DX57_LUPLU	23.0/40.9
Saccharolobus solfataricus	Not available	Not available
Streptomyces albus	Not available	Not available
Streptomyces flocculus	Not available	Not available
Thermococcus kodakarensis	Not available	Not available
Thermotoga maritima	MTNN_THEMA	27.2/46.6

3. Most SAHases of bacterial and eukaryotic origin require a monovalent cation (preferably potassium) for catalytic activity. Does the same apply to archaeal enzymes? Could the authors give some comments on that?

→ Judging from the structural data, our investigated archaeal SAHs do not have monovalent cations in the active site, suggesting that they are alkali-metal-independent, as has been also described for SAHs from cyanobacteria. We added this information together with corresponding references in the manuscript (see page MS page 13 (Line 10-21), “Differences in archaeal compared to bacterial and eukaryotic SAHs).

4. Are there any significant geometrical differences between the macromolecular environment of the substrate binding pocket of MmSAHH complexed with adenosine and inosine?

→ Our analysis of the macromolecular environment revealed no substantial geometric differences between the two complexes. The comparison of the MmSAHH•adenosine and MmSAHH•inosine structure shows a high degree of structural similarity in the substrate binding pockets, with conserved interactions between ligands and surrounding amino acid residues. We included the structures of the active site of MmSAHH with both substrates bound (see MS page 11, Line 8-11 with “Analysis of superposed structures of MmSAHH with bound adenosine53 and bound inosine showed no substantial differences in the architecture of the substrate binding pocket depending on the bound substrate” and SI page S44, Figure S28).

Figure S28. Superposition of the active sites of *MmSAHH* with adenosine (ADO) in orange (PDB ID: 5AXA) and with inosine (INO) in blue (PDB ID: 8COD) along with NAD⁺ cofactors and Na⁺ ions (SBP: substrate binding pocket; CBD: Cofactor binding pocket).

5. A figure of superposed substrate binding pockets of SAHases of bacterial, eukaryotic and archaeal SAHases would be helpful to illustrate significant differences in ligand binding.

→ Thanks for the suggestion, we included a new figure in our revised manuscript depicting the overlap of substrate binding pockets in the SAHs from mouse, *Pseudomonas aeruginosa*, and *Pyrococcus furiosus* (see MS page 15, Line 1-7, Figure 7). The aligned structures highlight key residues and interactions that determine the specificity of ligand binding. This figure now provides a clear comparison of structural variations that contributes to our understanding of ligand binding diversity among SAHH families, supporting our conclusions on functional implications (see page 11, Line 8-11 and page 16, Line 2-11 starting with “While archaeal enzymes deriving from the Crenarchaeota phylum (*SacSAHH* and *SsoSAHH*) show catalytic activity only with SAH, homologues from *T. maritima* and the euryarchaeal classes of *Methanococci* and *Thermococci* (*MiSAHH*, *MjSIHH*, *PfuSAHH*, *TkSAHH*, and *MmaSAHH*) show higher catalytic activity with SIH”, (Line 2).

Figure 7: Overall structural comparison of euryarchaeal PfuSAHH with SAHs from other domains. A: Superposition of the active sites of SAHs from bacterial (yellow), eukaryotic (blue) and archaeal (white) organisms with bound substrates and NAD⁺ cofactors (SBP: substrate binding pocket; CBP: cofactor binding pocket). B: SBP of bacterial PaSAHH in complex with adenosine (AND). Zn²⁺ and K⁺ are shown as yellow spheres (PDB ID: 6F3N34). C: SBP of archaeal PfuSAHH in complex with SIH (PDB ID: 7R38). D: SBP of eukaryotic MmSAHH in complex with inosine (INO). Na⁺ is shown as blue sphere (PDB ID: 8COD). E: In PfuSAHH, the cofactors are loosely covered by the adjacent monomer while some room is visible between the monomers, indicating a less stable enzyme (PDB ID: 7R38). F: In MmSAHH, the cofactors are tightly covered by the C-terminus of the adjacent monomer (PDB ID: 5AXA⁵³).

Reviewer 2

Popadic et al. deals with the structural and biochemical characteristics of S-adenosyl-L homocysteine hydrolases (SAHHs) and S-inosyl-L homocysteine hydrolases (SIHHs). These enzymes play a crucial role in the regulation of the methyl cycle within cells by hydrolyzing SAH (or SIH) into adenosine (or inosine) and L-homocysteine.

The manuscript presents crystal structures of SAHHs from archaeal sources and their complexes with cofactors. Comparative analysis of these structures with bacterial and eukaryotic SAHHs is also conducted.

This reviewer appreciates the paper's meticulous execution and technically sound structural and biochemical analyses. Having said that, this reviewer feels that this manuscript falls short in providing us of the understanding of the biological significance of these enzyme families. This reviewer also feels that the manuscript does not advance our understanding of this enzyme in the cellular metabolism of SAM, which governs various cellular phenomena.

This reviewer suggests submitting the manuscript to a more specialized journal, as Communication Biology may not be the most suitable venue for this study.

→ One goal of the revision was the streamlining and focusing of the manuscript. We hope that the main message is now clearer, highlighting the relevance of the SIH pathway, and the description of the fingerprint motif, as well as first structural information on archaeal SAHHs (please also see the replies to Reviewer 3's comments).

Reviewer 3

The introduction to the paper is exceptionally well written and was a pleasure to read. However, the main body of the paper requires editing to focus more clearly on the data being presented and to strip out excess detail. At some points the text felt like a combination of research paper and literature review, and it was difficult to tell which data were new and which were from the literature.

The main data presented are the crystal structures of three archaeal SAH hydrolase enzymes, plus the structure of the mouse SAH hydrolase in complex with inosine. Additionally, semi-quantitative enzyme assays were carried out on a large range of eukaryotic, bacterial and archaeal enzymes.

However, there is an incongruency between the presented sequence-based phylogenetic tree and the observed enzyme phenotypes, expressed here in terms of substrate preference, that is not satisfactorily resolved.

→ In order to test the suggested fingerprint motif for SIH-preferring SAHHs, we added five new enzymes to the group of SAHHs studied. This clearly confirmed the motif for SIH-preferring enzymes. In contrast, we could not identify a motif or a set of amino acid residues yet that allows to predict a general substantial acceptance of SIH (with SAH being the preferred substrate). This is for example the case for the SAHH from mouse. In our experiences, it is often the case, that it is not (yet) possible to find a clear explanation for substrates discrimination, due to a complex network of effects that is not obvious from sequence or structure alignments. We added some discussion on this to the manuscript (see MS page 10, Line 1-8 with "*The reason for substantial SIH degradation and synthesis by MmSAHH and PaSAHH remains yet unclear; likely, there will be other amino acid residues involved in substrate recognition which are not obvious from the sequence alignments, Line 1*"). Nevertheless, we believe that the identification of the SIH-preferring motif in combination with the relevance of the pathway via the deaminated compounds will

be a valuable starting point towards further research into alternative cofactor salvaging/synthesis pathways.

21.12.2023

The structures of the archaeal enzymes are described in great detail, as are the details of substrate binding. This information could have been much more succinctly described with a greater and more information-dense use of diagrams and data-summarising tables.

→ Thanks for the suggestion. We completely agree that the structure description was a bit excessive. We have taken the advice and summarised most of the data given in the text in a table (see SI page S5, Table S4: page S38, Table S8), and completely revised and streamlined the manuscript.

Table S4. Data collection and refinement statistics (Molecular Replacement).

	MmaSAHH • NAD•inosine	PfuSAHH • NAD•inosine	PfuSAHH • NAD•inosine (95 °C)	PfuSAHH • NAD•SIH	SacSAHH • NAD•adenosine	MmSAHH • NAD•inosine
PDB ID	7R3A	7R37	8QNO	7R38	7R39	8COD
Data collection						
Homodimer(s) per asymmetric unit	4	1	1	1	4	1
Space group	P2₁	P4₂2₁2	P4₂2₁2	P4₂2₁2	P1	I222
Cell dimensions						
a, b, c [Å]	65.8, 328.9, 85.1	112.5, 112.5, 122.8	111.7, 111.7, 122.1	111.7, 111.7, 121.6	84.4, 88.4, 138.6	98.2, 102.5, 173.4
α, β, γ [°]	90, 107.2, 90	90, 90, 90	90, 90, 90	90, 90, 90	78.9, 74.9, 64.9	90, 90, 90
Resolution [Å]	45.44–2.65 (2.74–2.65)	48.62–2.28 (2.37–2.28)	48.31–2.03 (2.11–2.03)	53.4–2.05 (2.12–2.05)	46.16–2.50 (2.59–2.50)	54.9–2.48 (2.56–2.48)
R _{merge}	0.1748 (1.075)	0.1495 (1.191)	0.2373 (2.425)	0.1048 (0.9204)	0.1348 (0.582)	0.177 (0.97)
I / σ	6.73 (2.13)	13.51 (1.66)	15.36 (1.54)	16.82 (2.70)	6.67 (2.35)	7.8 (2.1)
Completeness [%]	97.3 (98.2)	99.8 (98.8)	99.7 (97.6)	99.7 (99.9)	87.5 (85.9)	98.8 (99)
Redundancy	3.6 (3.5)	12.7 (8.3)	26.8 (26.6)	12.8 (13.4)	3.6 (3.6)	5.7 (5.9)
Refinement						
Search model (PDB ID)	1V8B ²³ Chain A	5AXA ³¹ Chain A	5AXA Chain A	5AXA Chain A	3H9U ³² Chain A	5AXA Chain A
Resolution [Å]	45.44–2.65 (2.74–2.65)	48.62–2.28 (2.37–2.28)	48.31–2.03 (2.11–2.03)	53.4–2.05 (2.12–2.05)	46.16–2.50 (2.59–2.50)	54.9–2.48 (2.56–2.48)
No. reflections	141280 (13525)	36333 (3519)	50113 (4818)	48676 (4780)	105739 (10341)	31048 (3056)
R _{work} / R _{free} [%]	18.8/24.3	16.1/21.6	15.5/20.3	17.6/21.9	22.9/28.2	17.0/21.8

No. atoms	26322	6891	7086	7147	27556	7067	21.12.2023
Protein	25440	6604	6660	6706	25760	6650	
Ligand/ion	608	126	126	88	504	128	
Water	274	161	300	353	1292	289	
B -factors	67.14	46.77	35.81	38.41	40.96	40.98	
Protein	67.59	46.98	35.67	38.16	40.23	41.16	
Ligand/ion	55.84	38.49	28.84	34.81	31.83	35.81	
Water	50.24	44.84	41.95	44.11	59.02	39.23	
R.m.s. deviations							
Bond lengths [Å]	0.030	0.014	0.009	0.014	0.008	0.014	
Bond angles [°]	2.33	1.81	1.48	1.84	1.69	1.86	

Each structure was solved from a single crystal. Values in parentheses are for highest-resolution shell.

Table S8 Interactions between the active site of SAHHs and the bound substrate.

Complex Type	Base moiety Interactions	Other Hydrogen Bonds	Non-bonded Contacts
SacSAHH•NAD•adenosine	Adenine: N1 - Thr53 N6 - Glu55, His344 N7 - His344	O2' - Glu147, Asp181 O3' - Lys177 O5' - His51, Asp124	Thr56, Thr148, His292, Leu338, Gly343, Met349, Phe353, NAD501
MmaSAHH•NAD•inosine	Hypoxanthine: N1 - Glu72 O6 - Lys74 N7 - His364	O2' - Lys198 O3' - Lys198 O4' - His70 O5' - His70, His313	Thr75, Asp143, Glu168, Leu355, Leu358, Gly363, Met369, Phe373, NAD502
PfuSAHH•NAD•inosine	Hypoxanthine: N1 - Glu57 O6 - Lys59 N7 - His350	O2' - Glu153, Lys183, Asp187 O3' - Thr154, Lys183 O5' - His55, Asp128, His298	Thr60, Leu341, Leu344, Gly349, Met355, Phe359, NAD601
PfuSAHH•NAD•SIH	Hypoxanthine: N1 - Glu57 O6 - Lys59 N7 - His350	O2' - Glu153, Asp187 O3' - Thr154, Lys183: SD - His55 N - Asp128, Ser79 O - Asn80 OXT - Phe299	Thr60, Gly297, His298, Leu341, Leu344, Gly349, Met355, Phe359, NAD502

MmSAHH•NAD•inosine

Hypoxanthine:
N1 - Thr57
N7 - His353

O3' - Thr157, Lys186
O4' - His55
O5' - His55, His301

Leu54, Glu59,
Thr60, Asp131,
Glu156, Asp190,
Leu347, Gly352,
Met358, Phe362,
NAD601

Overall, despite the large volume of carefully conducted work, the conclusions drawn seem rather underwhelming, and unfortunately are not likely to be of strong interest to those in the wider scientific community.

→ In our opinion, the conclusion is now much more focused, and especially the potential wider occurrence of natural cofactor analogues (also for other enzyme cofactors, see also doi: 10.1128/JB.00785-17, also included in the discussion) will be important for scientists working on physiological pathways as well as in vitro biocatalytic systems or the development of orthogonal pathways (please see also reply to comment of Reviewer 2).

REVIEWERS' COMMENTS:

Reviewer #1 (Remarks to the Author):

The revised version of the manuscript is improved, as the authors included all the advisable corrections. Therefore, the manuscript can be accepted for publication in the Communications Biology journal.

Reviewer #3 (Remarks to the Author):

The revised version of this paper is a considerable improvement on the original in terms of brevity and clarity. I think with these changes, the paper is suitable for publication. I have a few minor corrections as detailed below:

Lines 148-149:

"These findings strongly support the assumption of alternative routes for methyl metabolism coupled to purine salvage within some classes of Euryarchaeota and closely related bacteria."

should read

"These findings strongly support the assumption that alternative routes for methyl metabolism coupled to purine salvage exist within some classes of Euryarchaeota and closely related bacteria."

Fig 3a is not actually referred to in the text. This could be added in around line 123. Might be worth to point out that the tree (made from sequence similarity) is congruous with observed activity, but incongruous with phylogenetic relatedness of the organisms involved.

Line 198: "convey"? Do the authors mean "undertake"?

Line 201: "miss a 40 amino acid segment" should be "are missing a 40 amino acid segment"

Line 206: "conforming" should be "confirming"

Line 210: remove "matching"

Line 389: Are the ΔG int values quoted here really equivalent to "solvation free energy" values? I don't think they are the same thing.

29.02.2024

Point-to-point reply (manuscript ID: COMMSBIO-23-1804)

→ We thank both reviewers again for their time and constructive comments. Our replies to comments in BLUE

Reviewer 3

The revised version of this paper is a considerable improvement on the original in terms of brevity and clarity. I think with these changes, the paper is suitable for publication. I have a few minor corrections as detailed below:

Lines 148-149:

"These findings strongly support the assumption of alternative routes for methyl metabolism coupled to purine salvage within some classes of Euryarchaeota and closely related bacteria."

should read

"These findings strongly support the assumption that alternative routes for methyl metabolism coupled to purine salvage exist within some classes of Euryarchaeota and closely related bacteria."

→ The suggested change has been implemented (now lines 133-135).

Fig 3a is not actually referred to in the text. This could be added in around line 123. Might be worth to point out that the tree (made from sequence similarity) is congruous with observed activity, but incongruous with phylogenetic relatedness of the organisms involved.

→ Figure 3a is now referred to in lines 125-128: "The same results were observed for other representatives from Euryarchaeota (*MiSAHH*, *MmaSAHH*, *PfuSAHH*, and *TkSAHH*). As described before, the bacterial *TmSAHH*, which is closely related to these euryarchaeal enzymes (**Fig. 3a**), catalysed SAH and SIH cleavage and synthesis, in our hands also with a strong preference for the hypoxanthine derivatives."

→ We also adapted the Figure caption which now reads:

"Figure 3: Phylogenetic tree of investigated SAHs/SIHs with substrate range and used enzyme assays. a: The sequence-based phylogenetic tree

groups the tested SAHHs/SIHHs according to their substrate preference. Nevertheless, this does not fully correspond to the phylogenetic relatedness of the respective organisms: Representatives from Crenarchaeota exclusively accept SAH as substrate; in contrast, most family members tested from Euryarchaeota and a closely related thermophilic bacterium accept both SAH and SIH, including some homologues that clearly prefer SIH. Except for CgSAHH and LISAHH, bacterial and eukaryotic SAHHs accept both substrates with a substantial preference for SAH."

Line 198: "convey"? Do the authors mean "undertake"?

→ This has been corrected – we went for "conduct", however, we would be also happy with "undertake". Lines 170-171: "The differences in substrate range among the investigated SAHHs/SIHHs prompted us to **conduct** a detailed sequential and structural comparison to identify the underlying molecular basis."

Line 201: "miss a 40 amino acid segment" should be "are missing a 40 amino acid segment"

→ This has been corrected: Lines 170-171: "A sequence alignment of the enzymes tested in this work (Supplementary Fig. S26 online), showed that *MmSAHH* as well as *PαSAHH* **are missing a 40 amino acid segment** in the catalytic domain as opposed to the other eukaryotic and bacterial representatives."

Line 206: "conforming" should be "confirming"

→ This has been corrected: Lines 177-179: "The majority of SAHHs/SIHHs analysed in this work containing a shortened C-terminus are (hyper-)thermophilic, **confirming** the assumption that the length of the C-terminus correlates with the thermophilicity of the enzymes."

Line 210: remove "matching"

→ This has been corrected: Lines 179-182: "Based on our sequence analyses, the fingerprint motif suggested to distinguish extremophile SAHHs from mesophilic enzymes can be further specified: Crenarchaeota and a large part of Euryarchaeota feature HxTxE as a signature, this matches the sequence signature of bacterial and eukaryotic representatives (HxTxE(Q))."

Line 389: Are the delta G^{int} values quoted here really equivalent to "solvation free energy" values? I don't think they are the same thing.

→ We used the ΔG^{int} values as they are described in the PDBePISA tutorial.

→ This has been clarified in the text: Lines 331-333: "The prediction analysis of *PfuSAHH* tetramers (both inosine and SIH complexes) showed an average value for **the solvation free energy on formation of the assembly (ΔG^{int})** of $-129.0 \text{ kcal mol}^{-1}$."